# Multi-omics profiling of younger Asian breast cancers reveals distinctive molecular signatures

Zhengyan Kan [1], Ying Ding[1], Jinho Kim[2], Hae Hyun Jung[3], Woosung Chung[2], Samir Lal[1], Soonweng Cho[1], Julio Fernandez-Banet[1], Se Kyung Lee[4], Seok Won Kim[4], Jeong Eon Lee[4], Yoon-La Choi [5], Shibing Deng[1], Ji-Yeon Kim[6], Jin Seok Ahn[6], Ying Sha[1], Xinmeng Jasmine Mu[1], Jae-Yong Nam[2], Young-Hyuck Im[3,6], Soohyeon Lee[7], Woong-Yang Park[2], Seok Jin Nam[4] & Yeon Hee Park[3,6]

Breast cancer (BC) in the Asia Pacific regions is enriched in younger patients and rapidly rising in incidence yet its molecular bases remain poorly characterized. Here we analyze the whole exomes and transcriptomes of 187 primary tumors from a Korean BC cohort (SMC) enriched in pre-menopausal patients and perform systematic comparison with a primarily Caucasian and post-menopausal BC cohort (TCGA). SMC harbors higher proportions of HER2+ and Luminal B subtypes, lower proportion of Luminal A with decreased *ESR1* expression compared to TCGA. We also observe increased mutation prevalence affecting *BRCA1*, *BRCA2*, and *TP53* in SMC with an enrichment of a mutation signature linked to homologous recombination repair deficiency in TNBC. Finally, virtual microdissection and multivariate analyses reveal that Korean BC status is independently associated with increased TIL and decreased TGF-β signaling expression signatures, suggesting that younger Asian BCs harbor more immune-active microenvironment than western BCs.

[1] Pfizer Oncology Research, San Diego, CA 92121, USA. [2] Samsung Genome Institute, Biomedical Research Institute, Samsung Medical Center, Sungkyunkwan University School of Medicine, Seoul 06351, Korea. [3] Samsung Biological Research Institute, Samsung Medical Center, Sungkyunkwan University School of Medicine, Seoul 06351, Korea. [4] Department of Surgery, Samsung Medical Center, Sungkyunkwan University School of Medicine, Seoul 06351, Korea. [5] Department of Pathology and Translational Genomics, Samsung Medical Center, Sungkyunkwan University School of Medicine, Seoul 06351, Korea. [6] Division of Hematology-Oncology, Department of Medicine, Samsung Medical Center, Sungkyunkwan University School of Medicine, Seoul 06351, Korea. [7] Pfizer Oncology, Seoul 04631, Korea. These authors contributed equally: Zhengyan Kan, Ying Ding, Jinho Kim. These authors jointly supervised this work: Zhengyan Kan, Soohyeon Lee, Woong-Yang Park, Seok Jin Nam, Yeon Hee Park. Correspondence and requests for materials should be addressed to Z.K. (email: zhengyan.kan@pfizer.com) or to S.J.N. (email: sjnam@skku.edu) or to Y.H.P. (email: yhparkhmo@skku.edu)

Breast cancer (BC) remains a leading cause of cancer deaths despite recent progress in its prevention and treatment[1]. There are also wide variations in both incidence and mortality rates around the world as country-specific trends vary widely and may differ from global trends[2,3]. In countries of the Asia-Pacific region, rapid rises of BC incidence in recent years[1–3] have brought increased appreciation of Asian breast cancer as a distinct patient population. Most notably, peak age of Asian BC is much younger than that in western countries such as the United States, as approximately half of the Asian BC patients were pre-menopausal whereas only 15−30% of western BCs are pre-menopausal[4,5]. The distinctive demographics of Asian BC raised the issue of how to appropriately adapt therapeutic strategies mainly established in the western countries in Asia-Pacific countries such as Korea. Breast cancers arising in younger patients (YBC) are known to be more aggressive with increased risk of relapse and mortality[6]. In particular, YBCs with estrogen receptor-positive (ER+) diseases tend to be resistant to endocrine therapies such as tamoxifen compared to older patients[4]. A recent paper also suggests that young age was associated with significantly increased risk of mortality only among the Luminal subtypes[7]. There appeared to be higher proportions of triple negative/basal-like and HER2 subtypes but lower proportions of ER+/Luminal A subtypes in YBC than in older breast cancers (OBC). Hence, it remains controversial whether YBC has a unique biology or is only surrogate for aggressive intrinsic subtypes. The main cause of worse outcomes in YBC also remains undetermined[8].

Genomic and molecular profiling studies have significantly advanced our understanding of breast cancer biology along with increasing elucidation of its intrinsic molecular subtypes and genetic driver mechanisms[9–13]. Several studies have compared the molecular landscape of YBC with that of OBC using primarily Caucasian BC cohorts. ESR1 gene expression was reported to be significantly lower along with lower protein expression (IHC) and more hyper-methylation in YBCs than in OBCs[8,14]. GATA3 mutations were found to be significantly enriched in YBCs[15] while CDH1 mutations are enriched in OBCs[14]. Within the ER+ subtype, YBCs were reported to harbor elevated integrin/laminin, EGFR signaling, and TGF-β signaling expression signatures[14]. Higher expression signatures of proliferation, stem cell, and endocrine resistance were also associated with YBCs[15]. Recent studies comparing breast cancers from different racial groups have reported genomic differences mainly in the distribution of intrinsic molecular subtypes[16–18]. One study compared gene expression profiles of different age groups within a BC cohort of 113 Middle Eastern women and identified 63 genes specific to tumors in young women[19]. Another study compared gene expression and microRNA profiles between Chinese and Italian BCs and found lower prevalence of Luminal A subtype among Chinese BCs[20]. However, none of the studies to date conducted multi-omics profiling encompassing both the genome and the transcriptome of younger, pre-menopausal Asian BCs. It remains unclear what molecular differences distinguish younger Asian BCs from BCs in the western countries, the focus for most of the currently available genomic and molecular profiling work.

Here we report a study where we perform whole exome and transcriptome profiling of a large cohort of Asian BCs enriched in younger pre-menopausal patients. We then systematically compare different categories of molecular characteristics between our cohort and a benchmark BC cohort and are able to identify significant differences in molecular subtype distribution, mutation prevalence affecting oncogenes, and mutation and gene expression signatures.

## Results

**Overview of multi-omics profiling data.** We performed whole exome sequencing (WES) on 186 tumors and matched normal samples and transcriptome sequencing (RNA-Seq) on 168 tumor and 10 adjacent normal samples from a breast cancer cohort assembled by the Samsung Medical Center in Korea—SMC (Supplementary Data 1). We predicted somatic mutations and copy number variations (CNVs) from the WES data and gene expression from RNA-Seq data using published bioinformatics tools (see Methods). As a benchmark of the currently available breast cancer genomics data, we used WES and RNA-Seq data provided by the TCGA BRCA study (TCGA) consisting of 1116 subjects from the United States[11]. The vast majority of the SMC cohort (88.2%, n = 165) are pre-menopausal while only 19 patients (10.2%) are post-menopausal (Fig. 1a). In contrast, only 23.5% of TCGA are pre-menopausal while 72.3% are post-menopausal. For comparison analyses, SMC was divided into two age groups—YBC (age ≤ 40, n = 125) and IBC (age > 40, n = 62). TCGA was divided into three age groups—YBC (age ≤ 40, n = 181), IBC (40 < age ≤ 60, n = 562) and OBC (age > 60, n = 535). The TCGA cohort also has more lobular carcinoma cases and higher tumor purity levels than SMC (Table 1, Supplementary Data 2).

**Molecular subtype classification and distribution.** We classified intrinsic molecular subtypes for the BC cohorts using three methods—ER and HER2 immunohistochemistry (IHC) analyses, gene expression classifier (PAM50) and a naïve molecular classifier (NMC) based on ESR1, PGR, ERBB2 gene expression and ERBB2 copy number data. Three sets of classification results were highly concordant, with 88% of SMC samples having matching classifications between IHC and PAM50, 92% between NMC and PAM50 and 89% between IHC and NMC. A consensus classification was derived based on three classifications and used in subsequent analyses (Supplementary Fig. 1). SMC had significantly higher proportion of the ER+/HER2+ subtype than TCGA (16.1 vs. 5.4%, logistic regression (LR): p = 1.5e-05) but lower proportion of the ER+ subtype than TCGA (53.6 vs. 72.8%, LR: p = 3.3e-06) (Fig. 1b, c, Supplementary Data 3). SMC also had significantly lower proportion of the Luminal A subtype than TCGA (28.3 vs. 43.7%, LR: p = 7.5e-4) and higher proportion of Luminal B (39.2 vs. 33.2%, LR: p = 0.136). In fact, the majority of Luminal cases in SMC were Luminal B (58%) whereas the majority of Luminal cases in TCGA were Luminal A (57%) (Fig. 1d). These observed differences in the distribution of estrogen receptor-positive subtypes were concordant between gene expression-based classifications and IHC-based classifications, and remain statistically significant after correcting for confounding effects of tumor stage and histology subtype (Supplementary Data 3). We then applied an integrative genomics approach to classify samples from two cohorts into ten IntClust subtypes using gene expression and copy number data[21,22]. Comparison of subtype distribution revealed a significant enrichment of IntClust5, a subtype known to harbor intermediate levels of genomic instability and an enrichment of HER2 amplifications[21], in SMC compared to TCGA (Fisher's exact test (FE): p = 0.02). The "CNA-void" IntClust4 subtype characterized by a flat copy number landscape was enriched in TCGA (FE: p = 0.13) (Supplementary Fig. 2).

**Germline pathogenic mutations in BC predisposition genes.** Hereditary factors such as germline deleterious mutations in BRCA1 and BRCA2 genes significantly increases predisposition to breast cancer. About 1−5% of BCs were attributed to BRCA1/ BRCA2 mutations in primarily Caucasian BC populations[23] while

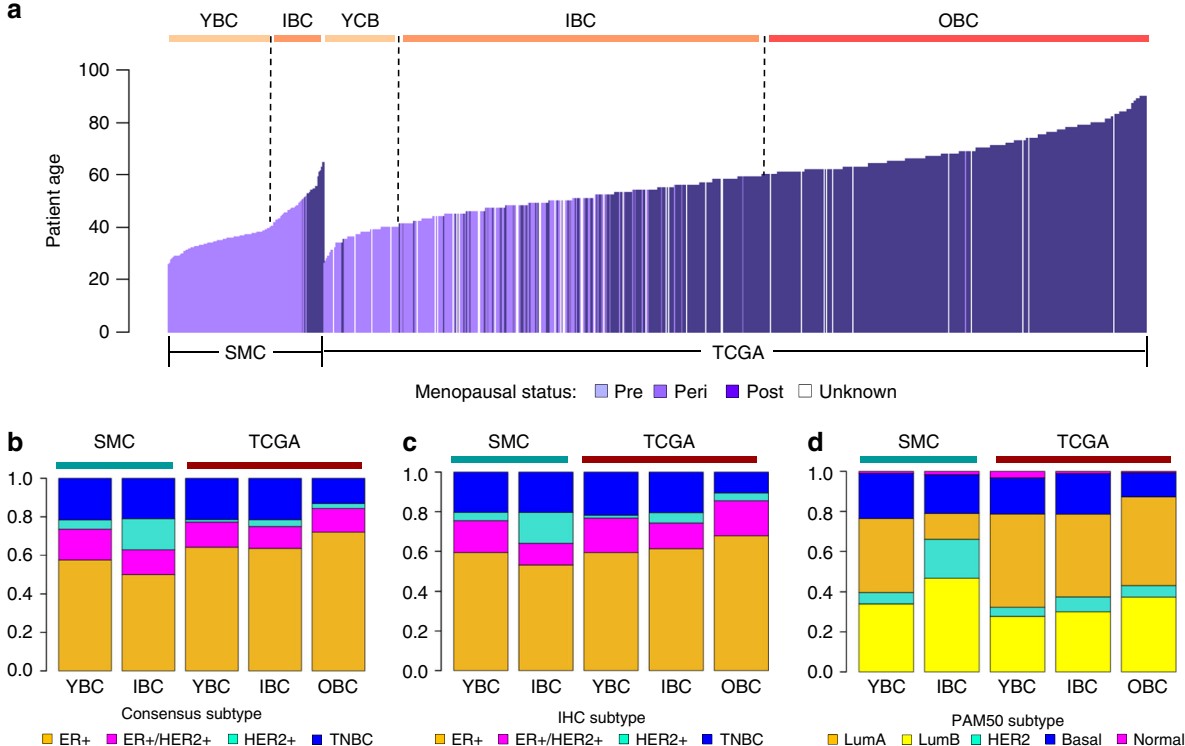

**Fig. 1** Age and molecular subtype distribution in SMC and TCGA. **a** Bar charts comparing the distribution of patient age between SMC and TCGA colored by menopausal status. Stacked bar charts comparing molecular subtype distribution across SMC and TCGA age groups based on three classifications: Consensus (**b**), IHC (**c**), and PAM50 (**d**)

3−7% of BCs were attributed to *BRCA1/BRCA2* mutations in Asian BC[24]. Higher prevalence of *BRCA1/BRCA2* mutation has been associated with younger age of diagnosis and family histories of breast cancers[23,24]. We examined germline pathogenic mutations, defined as mutations that truncate protein reading frame or reported as a disease-causing variant in ClinVar[25], in 13 genes known to increase breast cancer susceptibility with high to moderate penetrance[26]. In total, 18.8% (35/186) of SMC harbor germline pathogenic mutations in at least one of the 13 selected BC predisposition genes (Supplementary Fig. 3a). We found that *BRCA1* or *BRCA2* germline pathogenic mutations were significantly enriched in SMC compared to TCGA, affecting 10.8% of SMC but only 4.7% of TCGA (LR: $p = 0.0027$) (Supplementary Data 4−5). *BRCA1* or *BRCA2* germline mutations were enriched in younger patients, affecting 13.7% of SMC YBCs but only 4.8% of SMC IBC ($p = 0.08$) and 3.4% of TCGA OBC ($p = 6.85e-05$). Within TCGA, the YBC group also harbored higher prevalence of *BRCA1/BRCA2* mutations (12.0%) than IBCs (4.2%) and OBCs (3.4%) (Supplementary Fig. 3b). Moreover, SMC patients with family histories of breast cancers were significantly enriched in YBC relative to IBC (24 vs. 8%, FE: $p = 0.02$). Hence, *BRCA1/BRCA2* pathogenic mutations among other hereditary factors appeared to be a more prevalent cancer driver in this younger Asian BC cohort.

**Significantly mutated genes**. WES data analyses detected 6885 somatic protein-altering mutations affecting 4949 genes with an average of 0.6 missense mutations per Mb per sample (Supplementary Data 6). Observed mutation burden is significantly higher in TCGA (1.4 ± 4.5) than in SMC (0.90 ± 0.97) (Student's *t* test (ST): $p = 0.01$), consistent with recent reports that somatic mutation burden increases with age in cancers[27,28]. MutSigCV analysis[29] further identified six significantly mutated genes (FDR < 0.1) with a higher prevalence of somatic, protein-

altering mutations than expected—*TP53*, *PIK3CA*, *GATA3*, *CBFB*, *PTEN*, and *CDH1* (Table 2). All six genes were previously reported as being significantly mutated by the TCGA BRCA study, which also listed *TP53*, *PIK3CA*, and *GATA3* as top three mutated genes[11]. We also performed MutSigCV analysis on the combined list of mutations from two cohorts and identified 109 significantly mutated genes, all of which were already identified to be significant in TCGA or rarely mutated in SMC (Supplementary Data 7). *TP53* somatic mutation prevalence within the SMC cohort substantially varied by subtypes, with the highest mutation prevalence observed in TNBC (87.5%) followed by HER2+ (75%), ER+/HER2+ (64.3%) and ER+ (23.5%) (Supplementary Data 5). *TP53* mutation prevalence was significantly higher in SMC compared to TCGA overall (47.9 vs. 32.0%, LR: $p = 5.0e-5$) and after excluding Lobular carcinoma cases (49.4 vs. 37.3%, LR: $p = 0.003$) (Fig. 2b). *GATA3*, a transcription factor implicated in estrogen signaling, and E-cadherin (*CDH1*) were exclusively mutated in hormone positive subtypes. Consistent with earlier reports[14,15], *GATA3* mutation prevalence was higher in SMC (12.4%) than in TCGA (9.1%) whereas *CDH1* was more frequently mutated in TCGA (10.1%) than in SMC (2.4%).

**Comparing the landscape of genomic alterations**. We compared gene-level prevalence of somatic alterations integrating mutations and CNVs between SMC and TCGA overall and within subtypes (Fig. 2a). Notably, known oncogenes such as *TP53* (SMC: 47.9% vs. TCGA: 32%) and *ERBB2* (20 vs. 9.1%) were enriched in somatic alterations in SMC compared to TCGA overall (Supplementary Data 8). We then examined the prevalence of alterations at the pathway level by grouping frequently altered genes into five interconnected oncogenic pathways—receptor tyrosine kinase (RTK) signaling, MAPK signaling, PI3K/AKT signaling, cell cycle checkpoints and epigenetic regulators

**Table 1 Clinical data summary for SMC and TCGA**

|  | SMC | TCGA | Statistical significance |
|---|---|---|---|
| Subjects (n) | 187 | 1,116 |  |
| Patient age (yr) | 39.3 ± 8.5 | 58.3 ± 13.2 | p < 2.2e-16 |
| Tumor purity (%) | 71 ± 17.8 | 77.7 ± 10.7 | p = 0.0005 |
| Menopausal status (n(%)) |  |  |  |
| Pre-menopausal | 165 (88.2%) | 232 (20.8%) | p = 1.12e-76 |
| Post-menopausal | 19 (10.2%) | 715 (64.1%) | p = 2.4e-68 |
| Peri-menopausal | | 41 (3.7%) |  |
| N/A | | 128 (11.5%) |  |
| Clinical subtype (n(%)) |  |  |  |
| ER+ | 103 (55.1%) | 480 (43.0%) | p = 0.04 |
| ER+/HER2+ | 27 (14.4%) | 116 (10.4%) |  |
| HER2+ | 15 (8%) | 32 (2.9%) | p = 0.07 |
| TNBC | 37 (19.8%) | 121 (10.8%) |  |
| N/A | 5 (2.7%) | 367 (32.9%) |  |
| TNM stage (n(%)) |  |  |  |
| I | 27 (14.4%) | 177 (15.9%) |  |
| II | 101 (54%) | 605 (54.2%) |  |
| III | 58 (31%) | 237 (21.2%) |  |
| V | 1 (0.5%) | 18 (1.6%) |  |
| N/A | | 79 (7.1%) |  |
| Histology subtype (n(%)) |  |  |  |
| Lobular carcinoma | 7 (3.7%) | 193 (17.3%) | p = 2.9e-08 |
| Ductal Carcinoma | 172 (92.0%) | 830 (74.4%) | p = 2.9e-08 |
| Others | 8 (4.3%) | 37 (3.3%) |  |
| N/A | | 56 (5.0%) |  |
| Race (n(%)) |  |  |  |
| Asian | 187 (100%) | 57 (5.1%) | p = 2.53e-193 |
| Black | | 158 (14.2%) |  |
| White | | 745 (66.8%) |  |
| N/A | | 156 (14%) |  |

For SMC vs. TCGA comparisons of continuous and categorical variables, p values were calculated using Student's t test and Fisher's exact test respectively

(Supplementary Fig. 4, Supplementary Data 9). Cell cycle, PI3K/AKT, and RTK pathways were the most frequently altered, affecting 56.5, 36.0, and 28.0% of the SMC cohort respectively compared to 46.7, 42.7, and 21.7% of TCGA. Individually altered at low frequencies, epigenetic regulator genes in aggregate were altered in about 10% of SMC and 15% of TCGA. Cell cycle checkpoint pathway alterations were significantly enriched in SMC compared to TCGA (LR: p = 0.019). The higher frequency of ERBB2 amplification is probably the underlying cause for the enrichment of ER+/HER2+ and HER2+ subtypes in SMC compared to TCGA. Furthermore, as TP53 is more frequently inactivated in Luminal B than in Luminal A[11], the enrichment of Luminal B vs. Luminal A in SMC could result from higher prevalence of TP53 mutations as well as ERBB2 amplifications (Fig. 2c). Hence somatic alterations affecting oncogenic pathways when co-occurring with aberrant ER expression could play a greater role in the pathogenesis of younger Asian BCs.

**Identification and comparison of mutation signatures.** Mutation signatures are characteristic mutation patterns defined by different types of DNA damage that occurred as a result of exogenous and endogenous mutagens as well as DNA repair or replicative mechanisms[30]. We quantified the contribution of 30 predefined mutation signatures[31] in single tumors and detected 10 mutation signatures in the combined SMC and TCGA cohorts. These signatures include age-related signature S1, homologous recombination repair deficiency (HRD)-related signature S3, and APOBEC enzyme-related signatures S2 and S13 (Fig. 3, Supplementary Fig. 5a−b). Six of the signatures (S1-3, S5-6 and S13) were also found by a previous mutation signature analysis of 560 BC whole genomes[13]. Different intrinsic subtypes exhibited distinctive patterns of mutation signatures that were consistent between SMC and TCGA (Supplementary Fig. 5a−b). APOBEC signatures were over-represented in HER2+ or ER+/HER2+ tumors while HRD signature was predominant in TNBC (Fig. 3a, b, Supplementary Data 10). Mutation burden was the most strongly correlated with APOBEC signatures S2 and S13 in non-TNBC subtypes and the HRD signature S3 in TNBC, indicating that distinctive mutagenic processes are active in different intrinsic subtypes (Supplementary Data 10). APOBEC signatures S2 and S13 were enriched in ERBB2 amplified tumors in both cohorts, confirming previous report based on analysis of TCGA data alone[32] (Fig. 3d, e). HRD signature S3 was enriched in tumors harboring BRCA1/BRCA2 pathogenic mutations in both cohorts as expected (Fig. 3c). Notably, HRD signature S3 was significantly enriched in the TNBC subtype of SMC compared to TCGA (Fig. 3f). As much as 85% of TNBC cases in SMC were HRD positive (S3 score > 0.2) compared to only 52% of TNBCs in TCGA (FE: p = 0.7e-4). The HRD signature S3 was significantly correlated with younger patient age in both cohorts (Pearson correlation = −0.18, p = 1.0e-9) and under-represented in TCGA OBC (Supplementary Fig. 5a). Further, we observed an enrichment of S3 scores in tumors harboring recurrent BRCA1/BRCA2 germline missense variants compared to wild types in SMC but not in TCGA (Supplementary Fig. 5c−d). Hence homologous recombination repair deficiency may contribute more frequently to the carcinogenesis of TNBCs in SMC compared to TCGA, potentially due to differences in patient age as well as genetic predisposition.

**Virtual microdissection analysis.** Bulk tumor is a complex mixture containing tumor cells, stroma, immune cells, and normal tissue. We applied a computational approach called non-negative matrix factorization (NMF) to perform a virtual microdissection, separating bulk tumor gene expression into factors representing distinct tissue compartments[33]. We assembled a compendium of RNA-Seq data consisting of 1,678 samples, including tumor and adjacent normal tissue samples from SMC and TCGA, BC cell lines (CCLE) and healthy breast tissue samples (GTEx) (Supplementary Fig. 6a). NMF analysis on this expression compendium identified nine factors attributed to four tissue compartments—tumor, stroma, tumor infiltrating leukocytes (TIL), and normal tissue compartments (Fig. 4a; Supplementary Fig. 6b). Tumor intrinsic factors exhibited differential enrichment across molecular subtypes, allowing further association of these factors with ER+ (F4, F7), HER2+ (F2) and TNBC (F13) subtypes (Fig. 4b). The exemplar genes of factor F9 were enriched in immune and inflammatory pathways (Supplementary Data 11). Moreover, its factor weight was strongly correlated with the CYT score[34] and a wide range of immune cell signatures (Fig. 4c, d). Based on these evidences, we attributed F9 to the TIL compartment. Surprisingly, we found that TIL factor was significantly enriched in SMC compared to TCGA (Student's t test (ST): p = 1.55e-06) (Fig. 5a). Among the many F9 associated genes overexpressed in SMC vs. TCGA were CD8A, a marker of the cytotoxic T lymphocytes and PD-L1, an important mediator of immune checkpoint and target of multiple cancer immunotherapies (Fig. 5b).

**Table 2 Significantly mutated genes in SMC**

| Gene symbol | Gene description | SMC | | | TCGA | | SMC vs. TCGA |
|---|---|---|---|---|---|---|---|
| | | Mut Freq (n=186) | p value | q value | Mut Freq (n=1001) | q value | p value |
| TP53 | tumor protein p53 | 89 (47.9%) | 0 | 0 | 320 (32.0%) | 0 | 2.89e-04 |
| PIK3CA | phosphatidylinositol-4,5-bisphosphate 3-kinase, catalytic subunit alpha | 53 (28.5%) | 0 | 0 | 320 (32.0%) | 2.09e-12 | 0.81 |
| GATA3 | GATA binding protein 3 | 23 (12.4%) | 0 | 0 | 91 (9.1%) | 0 | 0.49 |
| CBFB | core-binding factor, beta subunit | 5 (2.7%) | 0 | 0 | 21 (2.1%) | 7.85e-07 | 0.83 |
| PTEN | phosphatase and tensin homolog | 6 (3.2%) | 6.40e-06 | 0.015 | 44 (4.4%) | 0 | 0.83 |
| CDH1 | Cadherin-1 | 4 (2.2%) | 3.64e-05 | 0.076 | 115 (11.5%) | 8.64e-13 | 2.46e-04 |

SMC has 186 WES samples and 167 tumor samples with both WES and RNA-Seq data. TCGA has 1002 WES samples. Mut Freq: number of mutated samples (% of mutated samples). p value, q value: mutation significance reported by MutSigCV. SMC vs. TCGA: Mutation prevalence were compared between SMC and TCGA by Fisher's exact test to determine the p value

Two ER+ associated factors (F4, F7) were strongly enriched in estrogen signaling pathways with F7 enriched in Luminal A than in Luminal B (Supplementary Fig. 7a). F7 was also significantly overweight in TCGA than in SMC within the ER+ subtype (ST: $p = 1.49e-04$). Moreover, F7 factor weight appeared to increase with patient age, with YBCs harboring the lowest weights and TCGA OBCs harboring the highest weight (Supplementary Fig. 7b). We noticed that the estrogen receptor gene (ESR1) expression was also positively correlated with patient age. In both SMC and TCGA, ESR1 expression was substantially higher in post-menopausal vs. pre-menopausal cases. ER+ tumors also expressed ESR1 at substantially higher levels in the older TCGA cohort compared to SMC (Supplementary Fig. 7c−d). Hence, ER+/Luminal A tumors were less prevalent in SMC while harboring weaker ER expression level and ER expression signature compared to TCGA, suggesting that SMC tumors were less dependent on estrogen receptor-mediated signaling.

**Differential expression analyses.** The SMC cohort was predominantly pre-menopausal (88.2%) while TCGA was predominantly post-menopausal (72.3%) (Table 1). Several studies have compared gene expression profiles of younger, pre-menopausal BCs with older, post-menopausal BCs within TCGA and identified differential expression (DE) of pathways associated with proliferation, stem cell, and endocrine resistance[14,15]. However, previous expression comparisons did not take into consideration the heterogeneous tissue compartments within bulk tumors. To identify molecular differences between the SMC pre-menopausal BCs (SMC-Pre) and the TCGA post-menopausal BCs (TCGA-Post), we performed differential gene expression analysis between two groups of tumors while adjusting for the confounding effect of molecular subtype and tumor purity. A total of 827 DE genes, among which 570 were upregulated in SMC-Pre and 257 were upregulated in TCGA-Post, were found to be statistically significant (see Methods, FDR < 0.01, fold-change > 2 or < 0.5) (Supplementary Data 12). We then computed GSVA scores of 6475 known pathway genesets[35] and performed similar differential comparisons between SMC-Pre vs. TCGA-Post. We found that 123 statistically significant DE pathways were upregulated in SMC-Pre and 591 were upregulated in TCGA-Post (Supplementary Data 12, Supplementary Fig. 8a).

To enhance biological insights into the molecular differences and mitigate confounding effects of heterogeneous tumor composition, we attributed DE genes and pathways to different tissue compartments in bulk tumor through correlation analysis with NMF factors (Supplementary Fig. 8b, Methods). We were surprised to see that more of the molecular differences were attributed to the tumor microenvironment (TME) than to the tumor intrinsic compartment. About 51.6% (427/827) of DE genes were associated with TME and 13.2% (109/827) of individual DE genes were associated with tumor intrinsic compartment. As much as 73% (253/345) of the DE pathways unambiguously assigned to a tissue compartment were attributed to TME while only 18% (61/345) were attributed to the tumor intrinsic compartment. We attributed 31 DE pathways to cohort-specific factors that may not represent true biological differences. The strongest DE patterns in TME are dominated by immune- and inflammation-associated pathways. About 86% (91/106) of the TIL factor-associated pathways were upregulated in SMC-Pre compared to TCGA-Post. T-cell markers, cytokine signaling genes as well as immune-related pathways such as allograft rejection, interferon α and IL-2 signaling pathways and immune cell type signatures such as cytotoxic cells and NK cells were also upregulated in the SMC pre-menopausal group (Supplementary Fig. 8c). On the other hand, 96% (144/150) of tumor stroma-associated pathways, such as response to TGF-β1 signaling, and nearly all of the tumor intrinsic DE pathways (58/61) were upregulated in TCGA-Post compared to SMC-Pre (Supplementary Fig. 8c). Eight of the 26 top tumor intrinsic DE pathways were associated with ER+ factors, further supporting the hypothesis that ER signaling is a more prevalent driver in TCGA compared to SMC BCs (Supplementary Fig. 8d).

Differential expression analysis of SMC pre-menopausal BCs (SMC-Pre) and the TCGA pre-menopausal BCs (TCGA-Pre) yielded similar observations. Most of the expression differences, 54.3% (425/782) of DE genes and 79% (333/423) of DE pathways, were associated with TME whereas only 10.2% (80/782) of individual DE genes and 14% (61/423) of DE pathways were associated with tumor intrinsic compartment (Supplementary Fig. 9a, Supplementary Data 12). Mainly immune- and inflammation-associated pathways were upregulated in SMC pre-menopausal tumors compared to TCGA pre-menopausal tumors (Supplementary Fig. 9b). Both TCGA pre-menopausal and post-menopausal BCs appeared to upregulate TGF-β1 signaling and estrogen signaling pathways compared to SMC pre-menopausal BCs (Supplementary Fig. 9b−c). The TGF-β signaling pathway has pleiotropic functions regulating multiple cellular processes and is known to play immune suppressive roles[36]. Further analysis revealed that the TGF-β expression signature quantified using GSVA[37] as well as TGFB1 gene expression were strongly enriched in TCGA compared to SMC within multiple molecular subtypes, suggesting that TCGA tumors harbor a more immune suppressed microenvironment than SMC tumors (Fig. 5c, d).

**Multivariate analyses of distinctive features.** To identify causal factors contributing to the observed molecular differences

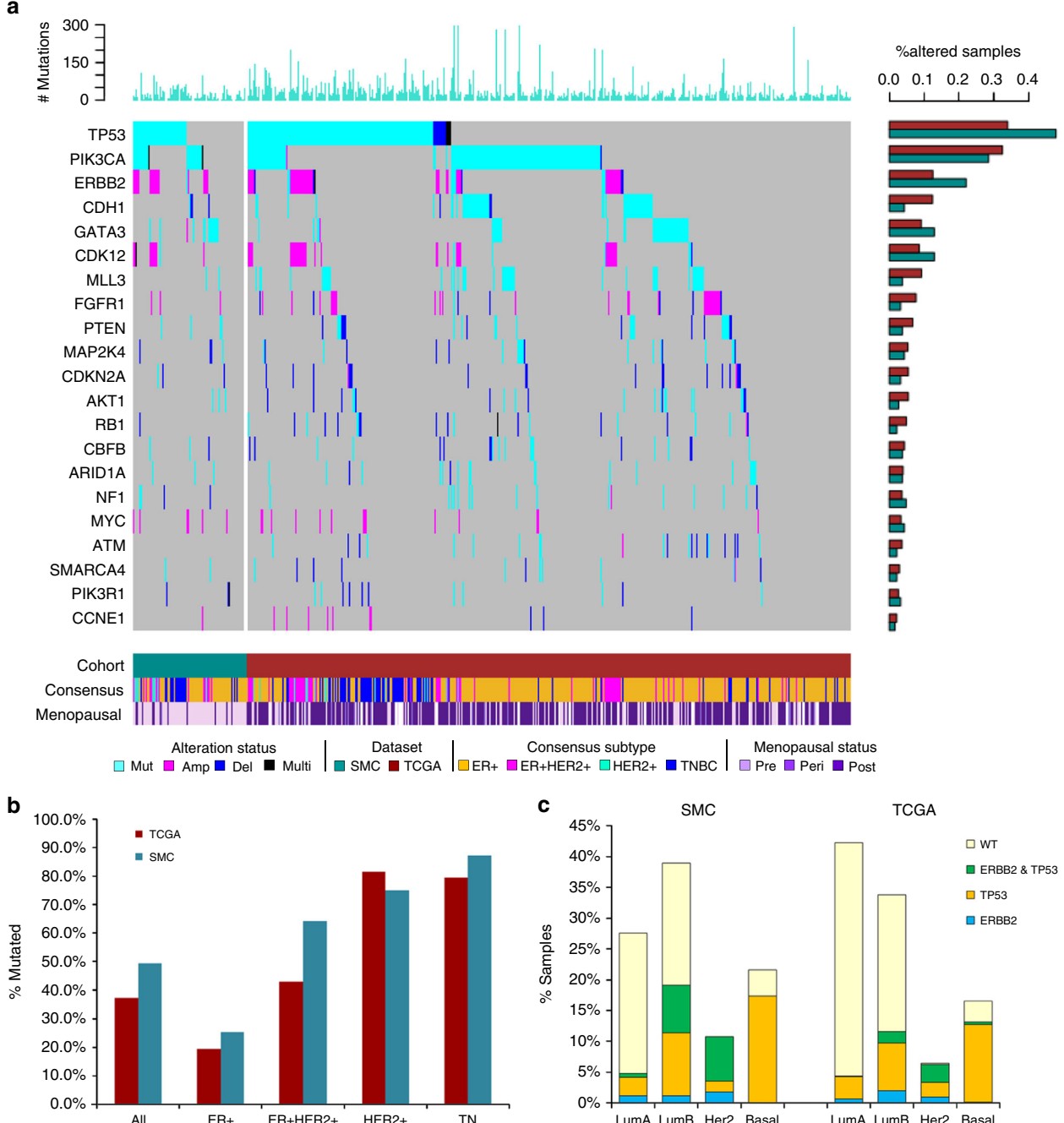

**Fig. 2** Landscape of somatic alterations in SMC and TCGA. **a** Heatmap showing the integrated statuses of protein-altering somatic mutations and copy number alterations affecting the most frequently altered genes (rows) and samples (columns) from SMC and TCGA. Cell colors represent different types of genomic alterations—Mut (mutation), Amp (amplification), Del (deletion), and Multi (multiple alterations). Genes are sorted by the prevalence of alterations (≥3 samples) in descending order. The bar chart above represents the sample-level count of protein-altering somatic mutations. The unstacked bar chart to the right compares the prevalence of somatic alterations in SMC and TCGA for individual genes. Column color labels represent age-based group (Cohort), intrinsic molecular subtype classifications (Consensus), and menopausal statuses for all samples. **b** Comparison of TP53 mutation frequencies between SMC and TCGA overall and within individual subtypes. Lobular carcinoma cases were excluded from frequency calculations. **c** Distribution of *TP53* mutation and *ERBB2* amplification statuses across PAM50 subtypes in SMC and TCGA. TP53: *TP53* mutated, ERBB2: *ERBB2* amplified (CN ≥ 6), ERBB2 & TP53: *TP53* mutated and *ERBB2* amplified, WT: wild-type *TP53* and *ERBB2*

between SMC and TCGA, we performed multivariate analyses to evaluate independent associations among distinctive molecular features and key clinical features (Fig. 5e). The Elastic Net method[38] was used to identify clinical and molecular features that were independently associated with the SMC vs. TCGA cohort status. For each feature, the variable usage statistic was determined based on the frequency of variable selection by the

predictive model and through bootstrapping[39] (Fig. 5e, Supplementary Data 13). The TGF-β signature and TIL factor F9 had the top variable usage of 100% among all distinctive molecular features. After adjusting for confounder effects of key clinical variables including patient age, tumor purity, molecular and histology subtypes, multiple regression analysis further demonstrated that both features were significantly and independently

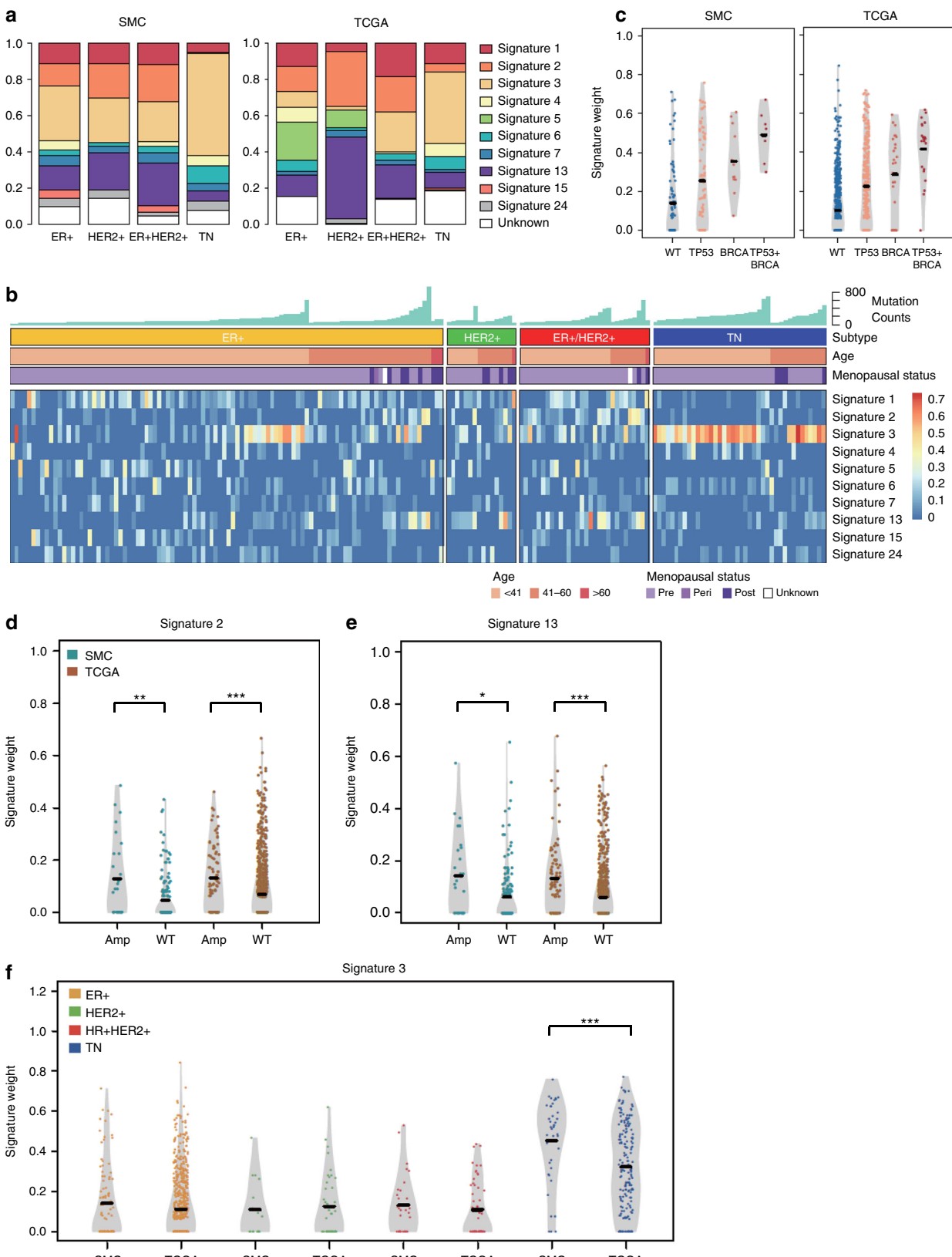

**Fig. 3** Landscape of mutation signatures in SMC and TCGA. **a** Distribution of major mutation signatures across subtypes in SMC and TCGA. **b** Heatmap showing scores of major mutation signatures in SMC samples grouped by molecular subtypes. Sample-level mutation counts, age group, and menopausal status are illustrated by colored column labels. **c** Violin plots comparing signature 3 scores in SMC and TCGA samples binned by *BRCA1/BRCA2* mutation and *TP53* mutation statuses. Horizontal line indicates the median value. Violin plots of APOBEC signatures S2 (**d**) and S13 (**e**) vs. *ERBB2* amplification status (amp: CN ≥ 6; WT: CN < 6) in SMC and TCGA; **f** Violin plots of HRD signature S3 vs. consensus molecular subtypes of SMC and TCGA. Student's *t* test: ***p < 0.001; **p < 0.01; *p < 0.05

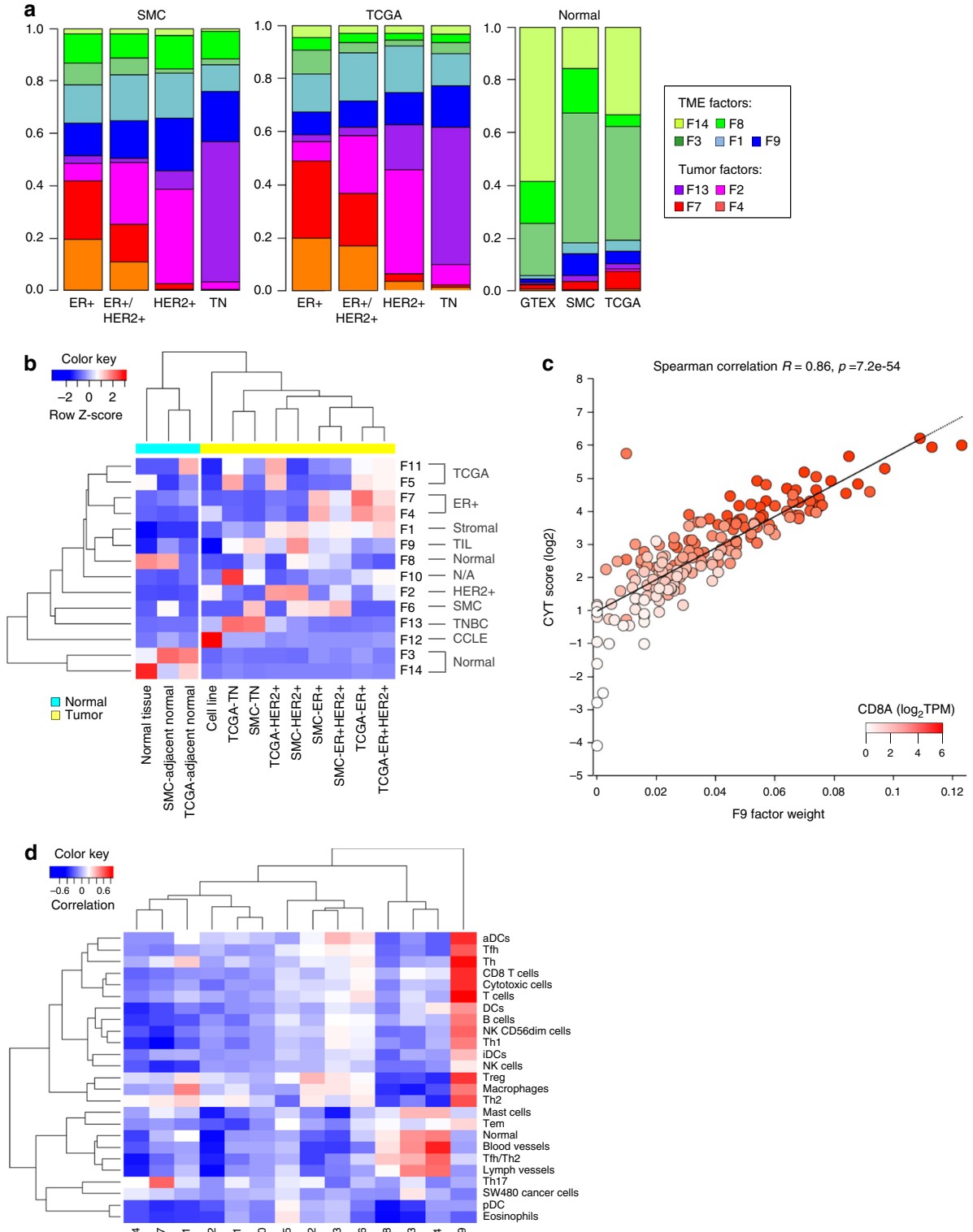

Fig. 4 Tumor virtual microdissection identified distinct tumor intrinsic and microenvironment factors. **a** Distribution of NMF factors representing TME or tumor intrinsic compartment across subtypes of SMC and TCGA. **b** Heatmap showing the mean sample factor weight for 14 NMF factors, z-normalized by rows, in different cohorts. Samples of normal tissue origins were clustered to the left and samples of tumor origins were clustered to the right. **c** Plot of CYT score vs. F9 factor weight with color gradient representing *CD8A* gene expression. **d** Heatmap of Pearson correlation coefficients between Bindea immune expression signature scores (GSVA) and NMF factors

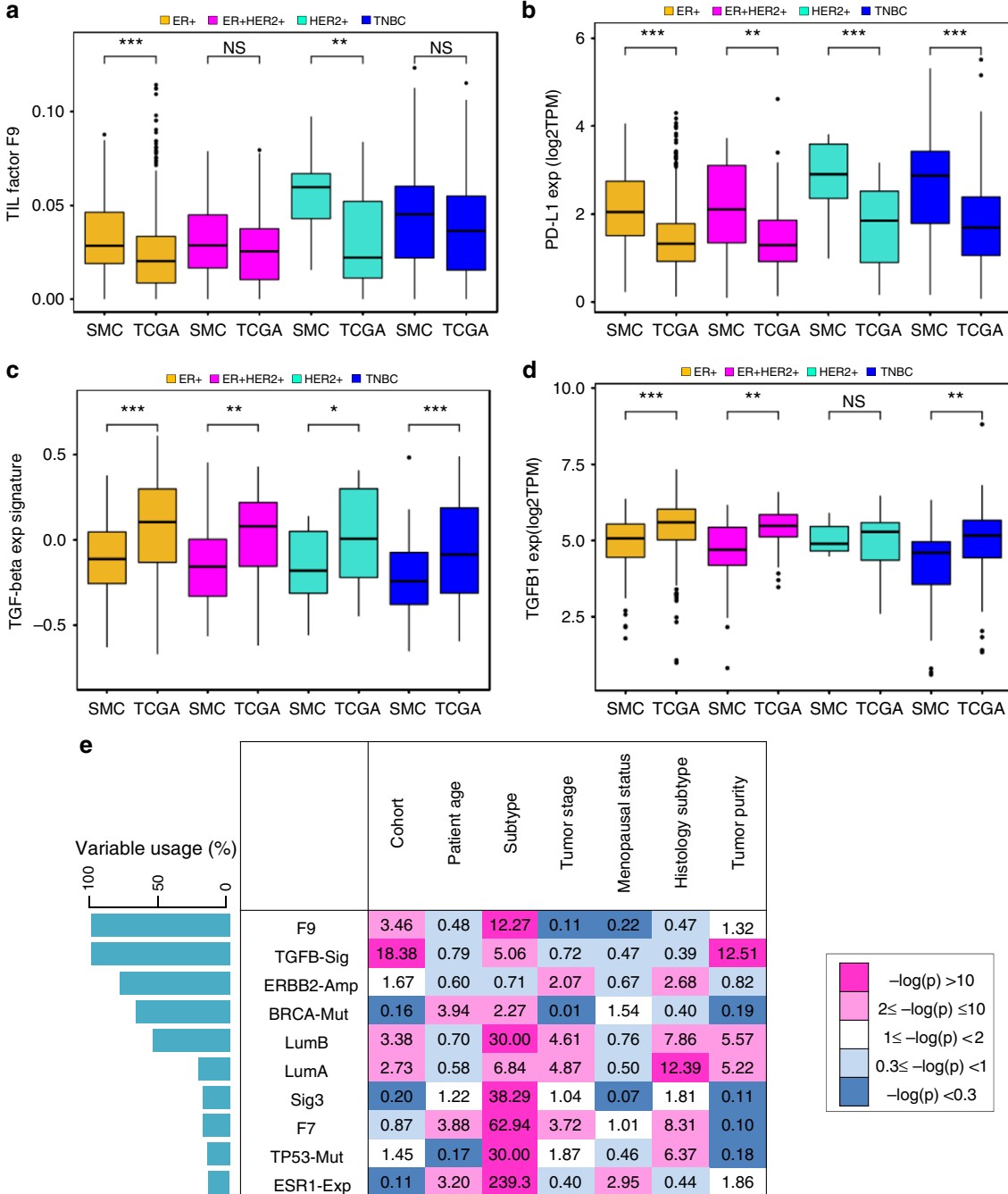

**Fig. 5** Distribution and multivariate analyses of distinctive molecular features. Boxplots showing the distributions of TIL factor F9 weights (**a**), PD-L1 gene expression (**b**), TGF-β expression signature score (**c**) and *TGFB1* gene expression (**d**) across molecular subtypes of SMC and TCGA. TPM transcript per million. *p*value was determined by Student's *t* test: ***$p < 0.001$; **$p < 0.01$; *$p < 0.05$; NS: $p \geq 0.05$. The box is bounded by the first and third quartile with a horizontal line at the median and whiskers extend to 1.5 times IQR. **e** Heatmap showing the statistical significance of associations ($-\log_{10}p$) between distinctive molecular features (rows) and key clinical features (columns) in the combined cohort. Molecular features were ranked by variable usage frequencies in descending order. Clinical features include two continuous variables—Patient age and Tumor purity and five discrete variables—Cohort status: SMC or TCGA; Subtype: ER positive or ER negative; Tumor stage: early (1−2) or late (3−4); Menopausal status: pre or post; Histology subtype: Lobular carcinoma or Invasive ductal carcinoma. Molecular features include S3: HRD-related mutation signature 3; F7: NMF factor 7 associated with ER+ subtype; F9: NMF factor 9 associated with TILs; ESR1-Exp: *ESR* gene expression; HER2-Amp: *ERBB2* amplification status; BRCA-Mut: *BRCA1/BRCA2* germline pathogenic mutation status; TP53-Mut: *TP53* somatic mutation status; TGFB-Sig: TGF-β pathway expression signature

associated with the cohort status (Supplementary Data 13). Hence, SMC BCs seemed to harbor a more immune active TME characterized by higher TIL factor while TCGA BCs harbored a more immune suppressed TME characterized by higher TGF-β signature, which were more likely to be due to ethnic or environmental factors rather than difference in age or menopausal status.

Regression analyses have also identified *ERBB2* amplification, Luminal A/B subtypes and TP53 mutation as having significant independent associations with the cohort status while *BRCA1/*

*BRCA2* pathogenic mutation and ER gene expression were independently associated with patient age (Fig. 5e). F7, one of the two NMF factors associated with ER signaling, was strongly associated with histology subtype (lobular carcinoma) as well as older patient age. *TP53* mutation remained significantly associated with the cohort status (multiple linear regression (MLR): $p = 0.035$) after adjusting for positive associations with intrinsic molecular subtypes and negative association with lobular carcinoma. HRD mutation signature S3 was strongly associated with *BRCA1/BRCA2* mutation (MLR: $p = 2.0e-08$) and TNBC (MLR: $p = 5.1e-39$) but not with cohort status. To further examine casual associations of S3, we performed multivariate analyses for only TNBC samples from SMC and TCGA. We observed that the variable usage frequency for S3 increased from 19.8% over all subtypes to 86.4% within TNBC, where independent associations were also increased for cohort status (MLR $p = 0.084$) and patient age (MLR $p = 0.036$) (Supplementary Data 11). Taken together, multivariate analyses have dissected the complex interrelationships among key clinical and molecular variables to reveal independent association of patient age with tumor intrinsic features such as HRD and ER signaling and link cohort status, a surrogate for genetic predisposition and environmental factors, to TME features such as TILs and TGF-β signaling.

## Discussion

In this study, we have performed whole exome and whole transcriptome profiling of 187 breast cancer tumors from a Korean cohort enriched in younger and pre-menopausal patients, portraying a distinctive patient segment that remained poorly characterized and underrepresented in previous genomic and molecular profiling studies. To identify distinguishing molecular characteristics of younger Asian BCs, we compared multi-omics profiles between our cohort and a BC cohort from the benchmark TCGA study, mainly consisting of Caucasian and post-menopausal patients. We observed that the landscapes of oncogenic alterations in SMC and TCGA bear the same hallmarks of cancer driver genes such as *TP53*, *PIK3CA*, and *GATA3* while *BRCA1* and *BRCA2* are the predominant BC predisposition genes in both cohorts. Upon closer examination, we found a number of significant differences in SMC compared to TCGA—lower proportions of ER+ and Luminal A subtypes but higher proportions of Luminal B, lower ER gene expression within ER+ subtypes, increased prevalence of *BRCA1/BRCA2* germline pathogenic mutations, *ERBB2* amplifications and *TP53* mutations, enrichment of HRD mutation signature (S3) in TNBC, increased levels of TIL factor (F9) and decreased levels of TGF-β signaling. Using multivariate analysis approaches, we sought to elucidate causal associations between these molecular distinctions and key clinicopathologic factors including cohort status and patient age. We found that tumor intrinsic molecular differences such as *BRCA1/BRCA2* mutation and ER signaling were mainly associated with patient age and menopausal status. On the other hand, TME-associated features such as TILs and TGF-β signaling were independently associated with cohort status after excluding the confounding effects of age, molecular subtype among other clinicopathologic features, pointing to genetic predisposition or environmental factors being the primary causes. Hence, younger Asian BCs appeared to harbor significant molecular differences from western BCs that could hold important implications for patient stratification and therapeutic treatment. As the SMC cohort was heavily enriched in younger patients and from a single institution, we believe our study provided a first step towards elucidating the molecular bases of Asian BCs which would require larger studies that include more patients from all age strata, multiple institutions and countries.

The apparent age disparity between the two BC cohorts was a major driver of observed molecular distinctions. *BRCA1/BRCA2* germline loss-of-function mutation frequencies were found to be age-dependent and enriched in SMC vs. TCGA, indicating there is a greater hereditary contribution to the pathogenesis of Asian BCs. Consistent with earlier reports, the SMC cohort harbored higher proportions of aggressive subtypes such as HER2+ and TNBC compared to TCGA. These younger BCs also expressed ER at lower levels along with weaker expression signature for ER signaling, suggesting that these tumors were less dependent on estrogen signaling than older BCs. Further, hormone receptor-positive tumors within SMC appeared to be more complex, harboring more Luminal B than Luminal A as well as a higher prevalence of oncogenic alterations including HER2 amplification and TP53 mutations. These findings in an Asian BC cohort independently corroborated and expanded previous reports of lower ER mRNA expression and higher HER2 mRNA expression in YBCs vs. OBCs[8]. Mutations affecting *TP53*, the most frequently mutated gene in cancer, are associated with more aggressive subtypes and resistance to chemotherapies[40,41]. Weaker tumor addiction to ER signaling coupled with co-occurring oncogenic drivers could in part explain the clinical observations that hormone receptor-positive YBCs respond more poorly to endocrine therapies than their older counterparts.

Mutation signatures are genomic footprints of mutagenic processes that occur throughout the lineage of the tumor cell and therefore can be used to infer cancer etiologies. We predicted the relative contributions of 30 pre-defined mutation signatures to the complement of somatic mutations in both SMC and TCGA BCs. Age-related signature S1, APOBEC signatures S2 and S13 and HRD signature S3 were the most predominant signatures identified. HRD signature was found to be significantly enriched in SMC compared to TCGA within TNBC, suggesting that DNA repair deficiencies may be a stronger etiological factor in younger Asian BCs. A class of inhibitors targeting the enzyme poly ADP ribose polymerase (PARP) has shown greater efficacies in treating tumors deficient in homologous recombination repair pathways, such as those harboring *BRCA1* or *BRCA2* mutations. Taken together with the observation of increased *BRCA1/BRCA2* mutation frequencies in younger BCs, our findings raised the possibility that PARP inhibitors could be more effective in treating younger Asian BCs particularly within the TNBC subtype.

Previous studies of younger vs. older breast cancers relied upon comparing gene expression profiles of primary tumors. As bulk tumor expression derive from a heterogeneous mixture of different cell types and tissue compartments, cohort-level comparisons tend to be confounded by differences in the composition of these mixtures. Virtual tumor microdissection through NMF analysis enabled us to separate expression differences that are tumor intrinsic from those attributed to other sources such as tumor stroma or even cohort-specific artifacts. Five of the 14 factors identified by NMF analysis were attributed to the TME but only four were attributed to tumor intrinsic biology. Differential expression analyses integrated with tissue compartment associations found that some DE features such as estrogen signaling are tumor intrinsic whereas the majority of DE features appear to reside in the TME. Moreover, SMC tumors appeared to harbor more inflammatory immune microenvironments than TCGA BCs, as indicated by higher expression levels of cytotoxic T-cell markers and checkpoint mediators such as PD-L1. At the same time, TCGA tumors appeared to harbor more immune suppressed TMEs that significantly upregulate *TGFB1* gene expression and TGF-β signaling compared to SMC BCs. Increased TILs have been associated with *BRCA1* mutation[42], younger patient age and ER-negative subtypes in breast cancers[43].

However, multivariate analyses indicated that the TIL factor was not independently associated with patient age or *BRCA1/BRCA2* mutation status. Hence, these observed differences in tumor immune microenvironment could represent an important distinction between Asian and western BCs inherently linked to hereditary and environmental factors. As immuno-oncology therapies are fast becoming a major addition to the anti-cancer arsenal, these findings call for greater strides in applying IO therapies as single agent or in combination with conventional therapies for the treatment of Asian BC patients.

## Methods

**Patient sample collection and multi-omics profiling**. This study was reviewed and approved by the Institutional Review Board (IRB) of Samsung Medical Center, Seoul, Korea (IRB No. 2013-04-005, 2012-08-065) with informed consents from the patients for the research use of clinical and genomic data. The tumor tissues and blood samples were prospectively collected at surgery. Ninety-five percent (179/187) of the tumors were treatment naive. All patients were diagnosed and treated at Samsung Medical Center. Patient samples were subjected to pathology review to ensure that >80% of samples are composed of ≥60% tumor cells (Supplementary Data 1). No statistical methods were used to predetermine sample size.

Genomic DNA from tumor was extracted using the QIAamp® DNA Mini kit (51304, Qiagen) and AllPrep DNA/RNA Mini kit (80204, Qiagen). Genomic DNA from whole blood was extracted using the QIAamp DNA Blood Maxi Kit (51194, Qiagen). Both tumor and blood DNA were enriched for exonic regions using the SureSelect XT regent kit (G9611B, Agilent) and SureSelect XT Human All Exon V5 kit (5190-6210, Agilent). Sequencing libraries were constructed for an Illumina HiSeq 2500 systems (Illumina) and sequenced in 100-bp paired-end mode of the TruSeq Rapid PE Cluster kit and TruSeq Rapid SBS kit (PE-402-4001, Illumina). Exome sequencing reads were mapped to the hg19 reference genome using Burrows-Wheeler Aligner (BWA-0.7.5a)[44]. PCR duplicates were removed by picard-1.93 (http://broadinstitute.github.io/picard). The mapped reads near putative indels were realigned and base quality was recalibrated using the GATK-2.4-7 suite[45]. We aimed for 100× mean target coverage for tumors and 50× for paired samples. For RNA-Seq, sequencing libraries were prepared using TruSeq RNA Sample Preparation kit v2 (RS-122-2001 and RS-122-2002, Illumina). Sequencing of the RNA libraries was performed on an Illumina HiSeq2500 in 100-bp paired-end mode of the TruSeq Rapid PE Cluster kit and the TruSeq Rapid SBS kit.

**Multi-omics data analysis**. RNA-Seq-derived gene expression data (TPM) from TCGA breast cancer, CCLE and GTEX normal breast samples were obtained from Omicsoft 2016 Q1 release, which consistently analyzed all datasets using the Omicsoft RSEM pipeline with the same library of gene models (Omicsoft-Gene20130723)[46]. Gene expression (TPM) was predicted from RNA-Seq data of 178 SMC samples (168 tumors, 10 normals) using RSEM[47], hg19 as the genome reference and the same gene model library as used by Omicsoft. We processed a subset of SMC RNA-Seq data using the Omicsoft RSEM pipeline and found an average correlation of 0.99 with expression profiles computed using the original RSEM, indicating that analysis results from two RSEM pipelines are highly compatible.

We obtained TCGA somatic mutation and CNV data from Omicsoft 2016 Q1 release, which derived the data from TCGA firehose release on January 28, 2016. Somatic mutations were detected from WES data on 186 SMC matched tumor/normal samples using Varscan2 v2.4.1[48] in the paired mutation calling mode. Parameters were selected based on the DREAM-3[49] setting of the false positive filter as recommended by Varscan2. All germline and somatic mutations were annotated using the Ensembl Variant Effect Predictor[50]. Significantly mutated gene analysis was performed using MutSigCV v1.2[29] from the online GenePattern tools[51]. We used GATK[52] to estimate depth of coverage from the bam files of 186 SMC tumor/normal samples using hg19 as genome reference. CNV segmentation was then called by ExomeCNV v1.4[53] based on coverage estimates and further annotated to derive copy number value for each gene.

**Molecular subtype classification**. We applied three methods for classifying molecular intrinsic subtypes—IHC, PAM50, and NMC. The IHC subtypes were determined by IHC assays for ER, PR, and HER2 and used in clinical diagnoses and treatment. PAM50 subtypes were predicted using Genefu[54] from the gene expression data. We developed an in-house breast cancer subtype classifier called naïve molecular classifier (NMC) based on *ERBB2*, *ESR1*, and *PGR* gene expression and *ERBB2* copy number. NMC assumed that HR-negative and -positive samples have expression values drawn from two Gaussian distributions with their respective mean and standard deviations. Using an expectation maximization (EM) method implemented by the mixtools package in R, we assigned probabilities of being drawn from either distribution to each sample with measured expression levels. Proportion of receptor positive/negative samples as measured by IHC were used as priors and EM algorithm was allowed to run until convergence. The distribution

with the higher probability was assigned as the NMC prediction for each of the receptors. We determined the Consensus subtypes by integrating classification results from IHC, PAM50, and NMC. If IHC is available, the majority vote by IHC, NMC, and PAM50 was used. Otherwise, the NMC classification result was used. If three methods disagree with each other, then the IHC result was used. The following rules were used to map IHC or NMC subtypes with PAM50 subtypes: ER+ (Luminal A and Luminal B), ER+ /HER2+(Luminal B and Her2), HER2+ (Her2) and TN (Basal).

**Integrated genomic subtype classification**. The iC10 package[22] in R was used for integrative cluster assignment for both the TCGA and SMC cohort. The classifier was trained on the genes using the "pamr" R package based on shrunken centroids. The optimal threshold for classification was determined using cross-validation. Intersecting genes from both gene-level copy number and gene expression data were used. The gene expression matrix was scaled to have a zero mean and standard deviation of 1 before integrative cluster assignment. Fisher's exact test and Chi-squared test were performed to assess significance of differences found in comparison analyses.

**Mutation signature analysis**. We obtained a predefined set of 30 mutational signatures from the Wellcome Trust Sanger Institute, which have been established by analyzing somatic mutation profiles of more than 10,000 tumor samples across 40 cancer types[31]. Each signature represents a characteristic pattern of 96 possible nucleotide substitution motifs, which combine six types of substitutions of central nucleotide and 16 combinations of the immediate flanking sequence. We included the complete set of the signatures in all the signature analyses as some signatures share similar patterns; thus exclusion of a signature may cause overestimation of the contribution of other signatures. To quantify the relative contribution of each mutational signature for tumor samples, we used deconstructSigs[55], which identifies the linear combination of input signatures to best explain the mutation profile provided in trinucleotide context. Once we computed sample-wise signature profiles, we filtered mutation signatures present in <20% of the samples in both SMC and TCGA cohorts. For group-wise comparisons, we pooled all the variants in a group-wise manner, and then applied deconstructSigs to quantify mutational signatures present in each group.

**Non-negative matrix factorization analysis**. The NMF algorithm factorizes the gene expression matrix V of g genes and s samples into two matrixes of k factors: gene factor matrix W of n gene weights for k factors and sample factor matrix H of m sample weights for k factors. W represents the expression pattern of the k parts and H represents the respective contribution of k parts in each sample or bulk tumor[56]. NMF was performed on log transformed gene expression matrix V, $\log_2(\text{TPM} + 1)$, of the combined cohorts using the R package NMF which used the "brunet" algorithm[57]. We performed 30 runs of NMF and chose the factorization that achieved the lowest approximation error for subsequent analyses. To extract exemplar genes for each of the k factors, a score for each gene g was first calculated representing how factor-specific it is based on an entropy measure[58]. Two criteria were then used for selecting the genes. First, the gene score has to be greater than $\bar{\mu} + 3\bar{\sigma}$ where $\bar{\mu}$ and $\bar{\sigma}$ represents the median and the median absolute deviation of the scores respectively. Second, the maximum contribution to a basis component of the feature has to be greater than the median of all contributions. Pathway enrichment analyses were performed on the exemplar genes for each factor using the Fisher's exact test and the MSigDB v5.1 pathway gene sets (Supplementary Data 11). Fractional contribution of each factor in each sample was calculated by normalizing W and H to facilitate cross factor comparison after excluding the cohort-specific factors. W was normalized so that $\sum_i w_{ij} = 1$, which is equivalent to multiplying a diagonal matrix S of k by k. Then H was multiplied by $S^{-1}$ and normalized by summing to 1. The average weights for each factor of various sample groups were calculated for Fig. 4.

To attribute NMF factors to different tissue compartments, we examined the distribution of sample factor weights in sample groups with known labels (Fig. 4a, b). For example, Factor F7 exhibits the highest sample weights for ER+ samples in SMC and TCGA, lower weights in other subtypes and the lowest weight in normal and cell line names. Based on this pattern of distribution, F7 was interpreted as a tumor intrinsic ER factor. Similarly we identified three additional tumor factors—ER factor (F4), HER2 factor (F2), and TN factor (F13)—and three factors associated with the normal tissues—F3, F8, and F14 (Supplementary Fig. 6). Factor F3 has higher weight in SMC or TCGA adjacent normal samples than in GTEx normal breast samples, indicating that it harbors biological characteristics specific to tumor adjacent normal tissue. In contrast, F8 and F14 have higher weights in GTEx normal breast samples than in SMC or TCGA adjacent normal samples, indicating that these factors are more presentative of health normal tissue. Besides the three normal factors, we also attributed two more factors to the TME. TIL factor F9 has the highest weights in tumor tissue groups such as SMC and TCGA, lower weights in adjacent normal groups and the lowest weights in GTEx normal and CCLE cancer cell line samples. F9 was also highly correlated with the CYT score, a measure of cellular cytolytic activities defined as the geometric mean of *GZMA* and *PRF1* expression levels[34] (Fig. 4c). F9 exemplar genes were significantly enriched in the immune- and inflammation-related pathways. Moreover, factor

9 sample weight is significantly correlated with expression signatures for many immune cell subtypes (Fig. 4d). The stroma factor F1 exhibited higher weights in SMC and TCGA tumor samples than in cancer cell line and GTEx normal samples. Moreover, F1 exemplar genes are significantly enriched in epithelial mesenchymal transition, extracellular matrix (ECM) organization and ECM regulation. Four factors (F5, F6, F11, and F12) appear to be cohort-specific factors due to significantly higher sample weights in one specific cohort—F5, F6 (TCGA), F6 (SMC), and F12 (CCLE). We did not find a clear association with any tissue compartment for F10, which did not yield any significantly enriched pathway.

**Differential expression and pathway enrichment analyses**. We filtered 25% of genes having the lowest mean expression levels and then 25% having the lowest variation measured as standard deviation before conducting differential expression analyses. Voom was applied to transform the gene-level normalized counts to log2-counts per million (logCPM). For the DE pathway analysis, the GSVA algorithm[37] was used to calculate signature scores for 6475 gene sets of collections H (hallmark gene sets), C2 (curated gene sets), C5 (GO gene sets), and C6 (oncogenic gene sets) from MsigDB v5.1[59]. The TGF-β pathway expression signature was based on the "HALLMARK_TGF_BETA_SIGNALING" gene set. Limma was then applied to perform the DE gene and pathway analyses and calculate the fold change differences and statistical significance[60]. Molecular subtype and tumor purity were adjusted as confounder variable in the linear model. DE pathways were selected if FDR < 0.01 and $|\log_2 FC| > 0.2$ (FC: fold-change).

**Association of DE genes and pathways with NMF factors**. Differential expression analyses were typically performed to compare two groups of bulk tumors without considering heterogeneity in tumor composition. What appears to be upregulation of a gene in one cohort vs. another may be due to tumor intrinsic upregulation of the DE gene or tumors from that cohort harboring greater proportions of non-tumor cells that overexpress the DE gene. In this study, we tried to identify compartmental origins of the differential expression patterns based on NMF inferred contribution of different tissue compartments in the bulk tumor. To associate DE genes with NMF factors, we calculated Pearson correlation $r$ between gene expression and factor sample weight across all samples of the expression compendium. The maximum correlation $\max(r)$ for each gene against all factors was used as the test statistics. Ten permutation runs were conducted on the gene expression matrix by reshuffling samples. The $\max(r)$ for all genes then formed the null distribution. For each DE gene, $p$ value was calculated based on the number of times the $\max(r)$ from the null distribution exceeded the test statistic and FDR corrected using the Benjamini–Hochberg method. A cutoff of FDR ≤ 0.05 was used to identify significant association with an NMF factor. Pearson correlation $r$ was also calculated between pathway signature scores and sample factor weight requiring $r ≥ 0.6$ to define an association. We then categorized DE pathways into four major types—tumor intrinsic, TME, cohort-specific, and ambiguous. For each pathway, the $\max(r)$ with categories of factors were calculated: $\max(r)_{tumor}$—tumor factors (F2, F4, F7, and F13); $\max(r)_{TME}$—TME factors (F1, F3, F8, F9, and F14); $\max(r)_{cohort}$—cohort-specific factors (F5, F6, F11, and F14). The following criteria were then applied: A pathway was classified as (1) tumor intrinsic if $\max(r)_{tumor} - \max(r)_{TME} > 0.2$ and $\max(r)_{tumor} > 0.2$; (2) TME if $\max(r)_{TME} - \max(r)_{tumor} > 0.2$ and $\max(r)_{TME} > 0.2$; (3) cohort-specific if $\max(r)_{cohort} - \max(\max(r)_{tumor}, \max(r)_{TME}) > 0.2$ and $\max(r)_{cohort} > 0.2$. Remaining DE pathways were classified as "ambiguous".

**Multivariate analyses of clinical and molecular features**. We performed logistic regression adjusting for confounder variables to compare gene-level prevalence of somatic and germline mutations among different groups of patients. The group status was set as the response variable with the case group as 1 and the control group as 0. The binary mutation status for each gene was designated as the main factor while other potential confounder variables were treated as covariates. The $p$ value associated with the main factor represented the statistical significance of the differences in mutation frequency between the case group and the control group. FDR was calculated using the Benjamini–Hochberg method.

We performed multiple regression analyses with adjustment for confounder variables to assess the associations between molecular differences and clinical features. For the combined TCGA and SMC cohort, seven clinical variables were evaluated—cohort status (SMC vs. TCGA), patient age, ER status (positive vs. negative), tumor stage (early vs. late), menopausal status (pre vs. post), tumor purity, and histology subtype (lobular carcinoma: yes vs. no). Nine molecular features were evaluated—HRD-related mutation signature 3 (S3), NMF factor 7 associated with ER+ subtype (F7), NMF factor 9 associated with TILs (F9), *ESR1* gene expression in log2TPM (ESR1-Exp), *ERBB2* amplification status (ERBB2-Amp), *BRCA1/BRCA2* germline pathogenic mutation status (BRCA-Mut) and *TP53* somatic mutation status (TP53-Mut). Logistic regression was applied if the molecular difference was a binary variable and regular linear regression was applied for continuous variables. For multiple linear regression analyses, we solved the function $Y = \beta_0 + \beta_1 X_1 + \beta_2 X_2 + \cdots + \beta_p X_p + \varepsilon$, where $\beta_j$ quantify the association between variable $j$ with the response. R function "lm" from "stats" package was used to estimate the regression coefficients $\beta_0, \beta_1, \ldots \beta_p$ and corresponding $p$ values. For logistic regression analyses, we solved the function $\log(p(x)/(1 - p(x))) = \beta_0 + \beta_1 X_1 + \beta_2 X_2 + \cdots + \beta_p X_p + \varepsilon$, where $p(x) =$

$\Pr(Y = 1|X)$ and $Y$ was the binary response variable. R function "glm" from "stats" package was used to estimate the regression coefficients $\beta_0, \beta_1, \ldots \beta_p$ and corresponding $p$ values.

Generalized linear model using penalization (Elastic Net[38]) was performed to evaluate independent association between each variable from the combined list of molecular and clinical features and cohort status (SMC vs. TCGA). Consistent lasso estimation and consistent selection of variables were achieved through bootstrapping. The % variable usage estimate was determined as the frequency of variable selection by the predictive model in 500 bootstraps. In addition, the statistical significance of association between each feature vs. cohort status was calculated using a generalized linear model that incorporated all clinical and molecular features as covariates.

**Data availability**. WES and RNA-Seq data are available at European Genome-Phenome Archive under the accession code EGAS00001002621 and GEO under the accession GSE113184. All other remaining data are available within the Article and Supplementary Files, or available from the authors upon request.

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

## Acknowledgements

This work was supported by a grant from the National Research Foundation of Korea (NRF-2017R1D1A1B03031497) and the Ministry of Health & Welfare Republic of Korea (HI13C2096). S.L. is funded by a fellowship from the Pfizer worldwide R&D postdoctoral program. We thank the TCGA, CCLE and GTEx consortia for making their data publicly available. Finally, we thank all members of the SMC-Pfizer breast cancer collaboration team for their contributions: Sripad Ram (Pfizer), Keith Ching (Pfizer), Eric Powell (Pfizer), Pamela Vizcarra (Pfizer), Paul Lira (Pfizer), Tim Nichols (Pfizer), Vinicius Bonato (Pfizer), Sooyoun Cho (SMC), Jonghan Yu (SMC), Eun-Suk Kang (SMC), Yuan-hua Ding (Pfizer), Jadwiga Bienkowska (Pfizer), and Paul Rejto (Pfizer).

## Author contributions

The study was initiated and designed by Y.H.P., S.H.L., and S.J.N. Collection of specimens was coordinated by S.K.L., S.W.K., and J.E.L. H.H.J., Y.H.P., J.-Y.K., J.S.A., and Y.H.-I. collected and organized clinical data. Y.-L.C. performed pathology review of tumor specimens. WES and RNA-Seq were done by W.C., J.K. and W.-Y.P. Data analyses were conducted by Y.D., Z.K., J.K., S.L., S.C., S.D., X.J.M., J.-Y.N., Y.S., and J.F.-B. Y.D., Z.K., S.L. and J.K. produced tables and figures. Z.K., Y.D., J.K., S.H.L., W.C., S.L., and Y.H.P. wrote the manuscript.

## Additional information

**Competing interests:** Z.K., Y.D., S.H.L., S.L., J.F.-B., S.C., S.D., X.J.M., and Y.S. were employees of Pfizer Inc. at the time the work was performed. The remaining authors declare no competing interests.

