## [Peer Review File · Nature Communications]

Reviewers' comments:

Reviewer #1 (Remarks to the Author):

In this paper Kan and colleagues present a valuable data source for 187 tumors from a Korean cohort of breast cancer patients. Whole exome and transcriptome profiles were provided with extensive analyses to classify tumor subtypes and to compare the landscape of genomic alterations. While the dataset generated for a large cohort of breast cancer patients is a significant contribution to the field, the main limitation is that this study is descriptive and lack of mechanistic insights. In addition, most findings have already been well-documented in the literature. Using virtual microdissection analysis, the authors attempted to arrive an interesting conclusion that tumor intrinsic and micro-environment factors may play important roles in SMC. Unfortunately, no experimental data was provided to support such a computational prediction.

Major concerns:

1. A number of conclusions in this study aim to demonstrate the significant differences between Korean breast cancer patients and a primarily Caucasian cohort (TCGA). It is important to ensure the 187 BC tumors are representative of Korean BC patients without sampling bias. In line 4-6 (Page 3), "approximately half of the Asian BC patients were pre-menopausal". In Table 1, 165 out of 187 (88.2%) Korean BC patients in this study were pre-menopausal. Does it mean that Korean BC patients differ significantly from the rest Asian BC patients?
2. The authors claimed an outstanding merit of this study is multi-omics profiling and integrative analyses. Surprisingly, minimum effort has been taken to establish the links between genomic alterations (mutations and CNVs) and the aberrations in gene expression.
3. If the primary goal of this study is to illustrate the relevance of race and ethnicity in the etiology of breast cancer, an informative comparison would be between pre-menopausal groups in SMC (165 cases) and TCGA (232 cases but may overlap with 57 Asian BC patients in the cohort). Although the comparison between SMC-pre and TCGA-post may yield some numbers with significant p-values (Page 7), the biological meaning of such a comparison is not clear.
4. One important issue in using tumor tissue for expression profiling is whether the tissues were collected prior to any chemo treatment. In addition, for the determination of genomic mutations or CNVs, it would be critical to know whether the patients have been subjected to drugs which may lead to DNA damage.

Reviewer #2 (Remarks to the Author):

The study by Kan et al performed an analysis of 178 tumor samples obtained from Korean women. 88% of the samples were from pre-menopausal women. The data was compared to the TCGA database and molecular profiles, mutation rates were evaluated. The major conclusion of the article was that Asian BC may harbor a more immune-active microenvironment than Western BCs. It is well written with a good description of findings and methods.

As authors state other studies have shown that younger breast cancer populations have increased proportion of BRCA mutations and TNBC. Both have been associated with increased immune cell infiltration into the tumor.

Also per Korean tumor registry over 50% of diagnosed patients are postmenopausal while this patient cohort is on 12% in that range, therefore a conclusion that Asian BCs are much different from western BCs might be overstated based on analysis of this present cohort.

Although authors attempted to control for the large differences in age and tumor type between the SMC and TCGA genomic data this certainly can introduce variability.

What are the major claims of the paper?

The major conclusion of the article was that Asian BC may harbor a more immune-active microenvironment than Western BCs.

As authors state other studies have shown that younger breast cancer populations have increased proportion of BRCA mutations and TNBC. Both have been associated with increased immune cell infiltration into the tumor. Premenopausal women have been noted to have lower rates of luminal A cancer in other studies [1, 2] and Younger women and TNBC have higher rates of TILs [3]. BRCA mutations are also associated with TILs [4] therefore their findings could be due to the cohort differences rather than intrinsic difference between Korean and western women. Although authors attempted to control for the large differences in age and tumor type between the SMC and TCGA genomic data this certainly can introduce variability.

Also, Korean women diagnosed with breast cancer, majority are postmenopausal based on national registry data [5-7], therefore the cohort on patients in this study is not truly representative of Korean BC and the conclusion may be too aggressively stated.

Overall, the data in the article is very interesting and identifying a significant population with DNA repair signature which might benefit from other treatment modalities like PARP inhibitors or platin containing chemotherapy is important for treatment considerations and clinical trial designs.

Also, identifying immune activation signature certainly may lead to novel therapy choices, identifying better biomarkers for response to immune mediated therapies and potentially pathways for resistance to these therapies.

Are they novel and will they be of interest to others in the community and the wider field?

The authors build on other studies and do a much more comprehensive study of both clinical, molecular and genomic differences in their patient populations.

Is the work convincing, and if not, what further evidence would be required to strengthen the conclusions?

It is difficult to assess the strength of the conclusion as the statistical methods for comparing of the SMA and TCGA are not described in great detail. But to truly show that Asian/Korean BCs are different from western BCs we would require data from a more representative cohort that included more postmenopausal women.

On a more subjective note, do you feel that the paper will influence thinking in the field?

The results of this paper might influence clinical trial designs to allow for stratification based on genomic/immune subtypes which may correlate with differential responses to therapies.

1. Elesawy, B.H., et al., Immunohistochemistry-based subtyping of breast carcinoma in Egyptian women: a clinicopathologic study on 125 patients. *Ann Diagn Pathol*, 2014. 18(1): p. 21-6.
2. Ithemelandu, C.U., et al., Molecular breast cancer subtypes in premenopausal and postmenopausal African-American women: age-specific prevalence and survival. *J Surg Res*, 2007. 143(1): p. 109-18.
3. Mahmoud, S.M., et al., Tumor-infiltrating CD8+ lymphocytes predict clinical outcome in breast cancer. *J Clin Oncol*, 2011. 29(15): p. 1949-55.
4. Nolan, E., et al., Combined immune checkpoint blockade as a therapeutic strategy for BRCA1-mutated breast cancer. *Sci Transl Med*, 2017. 9(393).
5. Min, S.Y., et al., The Basic Facts of Korean Breast Cancer in 2013: Results of a Nationwide Survey and Breast Cancer Registry Database. *J Breast Cancer*, 2016. 19(1): p. 1-7.
6. Kim, Z., et al., The Basic Facts of Korean Breast Cancer in 2012: Results from a Nationwide Survey and Breast Cancer Registry Database. *J Breast Cancer*, 2015. 18(2): p. 103-11.
7. Kim, Z., et al., The basic facts of korean breast cancer in 2011: results of a nationwide survey and breast cancer registry database. *J Breast Cancer*, 2014. 17(2): p. 99-106.

Reviewer #3 (Remarks to the Author):

This study by Kan and colleagues describes a genomic (exome/transcriptome) study of Korean breast cancer patients (178 cases), of which the majority (88%) are 'young' (pre-menopausal). Comparisons between the Korean cases and TCGA cases (which are largely post-menopausal) revealed differences in the prevalence of certain molecular subtypes and driver genes, such as higher HER2+, Luminal B, and p53 rates in the Korean cases. Notably, germline mutations in BRCA1 and BRCA2 were higher in the Korean population (10% vs 6.4%), and Korean cases had higher mutation signature proportions of homologous recombination deficiency (HRD). Expression based "virtual tissue decomposition" revealed higher levels of immune signatures in Korean populations, suggestive of a more immune-active environment. This study is of interest particularly because a significant number of breast cancer cases in Asia are younger in nature.

Major Comments

- 1) A major component of this manuscript involves comparison the profiles of the Korean tumors to TCGA. In order for robust comparisons, ideally both data sets should be remapped according to the same pipeline. From the Methods, it appears that there are differences in the data pre-processing (eg Firehose for TCGA, Varscan for Korea). The analysis should be repeated using a common pipeline.
- 2) Many of the differences between the Korean and the TCGA cases may be inter-related. For example, if there are more ER negative cases in the Korean set, then this may explain the higher frequency of TP53 mutations in the Korean set, and maybe even the differences in the microenvironment (eg more TP53 mutations -> more mutations -> more neoantigens -> more immune cells etc). I would like to see the authors attempt a multi-variate analysis that takes into account all the individual differences seen (eg age, immune cells etc) that are significant at the individual factor level, and assess which remain significant after multi-variate adjustment.
- 3) For the cases with germline BRCA1/2 mutations, how many experienced a second inactivating hit in the tumor (eg LOH, second mutation, silencing which could be investigated by looking at loss of expression, etc).
- 4) Have the others explored redoing mutation significance analysis on the combined TCGA and Korean cohort? More driver genes may be identified due to the expanded number of samples.
- 5) Was the F9 factor specific to a particular molecular subtypes of breast cancer (eg ER negative)?
- 6) Have the differences in the tumor microenvironment observed in the expression analysis been corrected for potential differences in tumor purity, which is higher in the TCGA cohort? It seems that if one were to compare high-tumor cellularity tumors to low tumor-cellularity cases, then it seems obvious that differences in tumor microenvironment genes will be seen. Is this true biology or an artefact of the sampling procedure?

Minor Comments

- 1) Table 1 should highlight which clinical factors are significantly difference between the Korea and TCGA cohorts.
- 2) Are the BRCA1/2 mutations seen in the Korean cases novel, or have any of these been previously reported in other studies (perhaps from Asian breast cancer families)?

Reviewers' comments

Reviewer #1 (Remarks to the Author):

In this paper Kan and colleagues present a valuable data source for 187 tumors from a Korean cohort of breast cancer patients. Whole exome and transcriptome profiles were provided with extensive analyses to classify tumor subtypes and to compare the landscape of genomic alterations. While the dataset generated for a large cohort of breast cancer patients is a significant contribution to the field, the main limitation is that this study is descriptive and lack of mechanistic insights. In addition, most findings have already been well-documented in the literature. Using virtual microdissection analysis, the authors attempted to arrive an interesting conclusion that tumor intrinsic and micro-environment factors may play important roles in SMC. Unfortunately, no experimental data was provided to support such a computational prediction.

Major concerns:

(1) A number of conclusions in this study aim to demonstrate the significant differences between Korean breast cancer patients and a primarily Caucasian cohort (TCGA). It is important to ensure the 187 BC tumors are representative of Korean BC patients without sampling bias. In line 4-6 (Page 3), “approximately half of the Asian BC patients were pre-menopausal”. In Table 1, 165 out of 187 (88.2%) Korean BC patients in this study were pre-menopausal. Does it mean that Korean BC patients differ significantly from the rest Asian BC patients?

We thank the reviewer for pointing out this important issue. We agree with the reviewer that this SMC cohort is younger in age than Asian/Korean BCs in general and therefore not representative of the age distribution for Asian BCs overall. We would like to clarify that the goal of this study was not to compare all Asian BCs vs. Western BCs, but to characterize the genomic and molecular profile for an important subset of Asian BC patients who are younger and pre-menopausal. This patient segment makes up a significant portion of Asian BCs but is under-represented in western BCs. As discussed in the INTRODUCTION, the molecular bases for this distinctive BC segment remained poorly characterized because previous initiatives such as TCGA and METABRIC mainly focused on western BCs. Hence, towards the goals of elucidating the molecular bases and improving treatment outcomes for all Asian BCs, we feel that our study provides an important first step to delineate the genomic/molecular landscape of younger Asian BCs.

The comprehensive multi-omics dataset from the TCGA BRCA study provided a valuable reference to the scientific community for studying the molecular bases of BCs. Comparison analyses of genomic and molecular profiles between SMC and TCGA served two purposes. First, it allowed us to place the newly generated SMC data into the context of what people already knew about the molecular bases of BC and identify potentially novel and meaningful insights into the disease biology of younger Asian BCs. Second, it shows that there exist significant molecular differences between younger Asian BCs and the reference BC molecular profiles, thus revealing gaps in the existing data/knowledge and emphasizing the need to perform thorough molecular characterization of BCs from different racial backgrounds and age strata. After the identification of significant molecular differences between two large cohorts that differed in ethnicity, age among other factors, we sought to decipher which factors were causal of the observed molecular distinctions. We believed it would be scientifically significant to identify molecular features associated with younger BCs as well as Asian BCs, as both patient segments remained poorly understood and under-represented in the currently available molecular data. Towards this end, we have applied rigorous multivariate analysis approaches to deconvolute the complex associations among key clinicopathologic factors and distinctive molecular features including enrichment of TILs, increased HRD in TNBC and down-regulated TGF- β and ER signaling.

Innovative analyses of virtual microdissection and integration with DE analyses further revealed tissue compartmental origins for these distinctive molecular features. We found that tumor microenvironment associated features such as TILs and TGF- β signaling were more strongly associated with cohort status than with patient age. On the other hand, tumor intrinsic molecular features such as *BRCA1/BRCA2* mutations and ER signaling were independently associated with patient age/menopausal status rather than with cohort status.

In summary, the main goal of this study was to study the molecular bases for younger Asian BCs, an important but poorly characterized patient segment distinguished by patient age and ethnicity from the established molecular profiles of BCs. By comparing the genomic and molecular profiles of SMC with TCGA, we found several distinctive molecular features of younger Asian BCs that hold potential clinical implications. Further, through applying rigorous and innovative analysis approaches, we have “dissected” the tumor tissue compartment origins and identified likely causal factors for these distinctive features.

Reviewer’s comment indicated to us that the previous draft was not sufficiently clear in explaining these points. We have made the following revisions to the manuscript to address this issue:

- We have revised the last paragraph in INTRODUCTION to emphasize that our goal was to study younger Asian BC.
- We have revised the first paragraph in DISCUSSION to explain the rationale and approach of this study. We also explicitly pointed out the limitations of this study in that our cohort is biased towards younger patients and therefore do not represent overall Asian BC population.
- We have revised ABSTRACT to discuss using multivariate analyses to separate the causal effects of patient age from cohort status i.e. Korean vs. American BC.
- We also revised various parts in the manuscript to avoid claiming that observed differences exist between all Asian BCs and western BCs. We noted that molecular differences were observed between two datasets, SMC vs. TCGA, and potentially represent distinctive molecular features of younger Asian BC populations. Only by applying multivariate analyses were we able to make inferences about which molecular distinctions may be specific to Asian BC in general.

(2) The authors claimed an outstanding merit of this study is multi-omics profiling and integrative analyses. Surprisingly, minimum effort has been taken to establish the links between genomic alterations (mutations and CNVs) and the aberrations in gene expression.

We appreciate the reviewer’s concern about a lack of integrative analysis between genomic alterations and gene expression. To address this concern, we have applied a validated integrative genomic approach (IntClust) established by the METABRIC study to classify breast cancer samples in SMC and TCGA into ten subtypes using gene expression and copy number data. We found that two subtypes, “HER2+ enriched” IC4 and “copy number flat” IC5, are differentially distributed in SMC vs. TCGA, but did not observe other significant associations for the IntClust subtypes.

The following modifications were made to the manuscript:

- Added Supplementary figure 2
- Added a paragraph to the RESULTS section “Molecular subtype classification and differential distribution” to describe analysis results.

We believe that the first priority for a multi-omics study is to perform a thorough analysis of individual alteration types such as somatic mutation before venturing into more complex and integrative analyses. Integrative analysis is a broad category of analyses that are often exploratory in nature. We have performed and described integrative analyses such as integrating tissue compartment inference with differential expression analysis of bulk tumor expression profiles. There were multiple other analyses that we performed but did not yield significant results suitable for including in this manuscript. As a next step, we agree that it will be important to perform more integrative analyses that could be reported in separate manuscripts.

(3) If the primary goal of this study is to illustrate the relevance of race and ethnicity in the etiology of breast cancer, an informative comparison would be between pre-menopausal groups in SMC (165 cases) and TCGA (232 cases but may overlap with 57 Asian BC patients in the cohort). Although the comparison between SMC-pre and TCGA-post may yield some numbers with significant p-values (Page 7), the biological meaning of such a comparison is not clear.

We thank the reviewer for a great suggestion. As most of the SMC BCs are pre-menopausal and most of the TCGA BCs are post-menopausal, the DE analysis of SMC-Pre vs. TCGA-Post served as the primary expression comparison analysis between two cohorts. There was also an interest in identifying molecular differences between younger vs. older breast cancers. We agree that it is difficult to interpret the biological meaning of the differences which may be related to a multitude of factors including ethnicity, age and menopausal status. A comparison between two subgroups having similar age range and menopausal status is more useful in focusing on molecular differences due to race/ethnicity.

We have performed DE comparison between SMC and TCGA pre-menopausal BCs as suggested. The analysis results were similar to DE comparison between SMC-Pre and TCGA-Post in that the majority of molecular differences were associated with the tumor microenvironment. Immune related pathways also dominated up-regulated pathways in SMC while TGF- β and estrogen signaling pathways were strongly up-regulated in TCGA. The following modifications were made to the manuscript to describe the analysis results:

- Supplementary Figure 9a: DE gene expression patterns.
- Supplementary Figures 9b: DE pathways associated with tumor microenvironment (b) and tumor intrinsic factors (c).
- Supplementary Table 12a-b: differentially expressed genes and pathways.
- Added a paragraph to the “Integrative analyses of differential expression and tumor composition” section in RESULTS.
- We placed more emphasis on reporting TGF- β signaling as a molecular distinction.

(4) One important issue in using tumor tissue for expression profiling is whether the tissues were collected prior to any chemo treatment. In addition, for the determination of genomic mutations or CNVs, it would be critical to know whether the patients have been subjected to drugs which may lead to DNA damage.

We agree with the reviewer that this is an important issue. The vast majority of the tumors from both cohorts were surgical biopsies and treatment naïve. Only 5% (9/187) of the samples in the SMC cohort and 1% (13/1,116) of the TCGA cohort were treated by neoadjuvant chemotherapies prior to tumor resection. We have added a statement in the METHODS section “Patient sample collection, whole exome and transcriptome sequencing” to point out this fact. We also added a column “chemotherapy_treated” in Supplementary Table 1 to indicate whether the tumor has received prior chemotherapy treatment.

Reviewer #2 (Remarks to the Author):

The study by Kan et al performed an analysis of 178 tumor samples obtained from Korean women. 88% of the samples were from pre-menopausal women. The data was compared to the TCGA database and molecular profiles, mutation rates were evaluated. The major conclusion of the article was that Asian BC may harbor a more immune-active microenvironment than Western BCs. It is well written with a good description of findings and methods. As authors state other studies have shown that younger breast cancer populations have increased proportion of BRCA mutations and TNBC. Both have been associated with increased immune cell infiltration into the tumor. Also per Korean tumor registry over 50% of diagnosed patients are postmenopausal while this patient cohort is on 12% in that range, therefore a conclusion that Asian BCs are much different from western BCs might be overstated based on analysis of this present cohort. Although authors attempted to control for the large differences in age and tumor type between the SMC and TCGA genomic data, this certainly can introduce variability.

What are the major claims of the paper?

The major conclusion of the article was that Asian BC may harbor a more immune-active microenvironment than Western BCs.

(1) As authors state other studies have shown that younger breast cancer populations have increased proportion of BRCA mutations and TNBC. Both have been associated with increased immune cell infiltration into the tumor. Premenopausal women have been noted to have lower rates of luminal A cancer in other studies [1, 2] and Younger women and TNBC have higher rates of TILs [3]. BRCA mutations are also associated with TILs [4] therefore their findings could be due to the cohort differences rather than intrinsic difference between Korean and western women. Although authors attempted to control for the large differences in age and tumor type between the SMC and TCGA genomic data, this certainly can introduce variability.

We thank the reviewer for pointing out a very important caveat and providing many helpful references. It is a great question whether the observed enrichment of TILs in SMC vs. TCGA is due to difference in cohort composition, for instance, SMC having more TNBCs and/or more *BRCA1/BRCA2* mutations than TCGA as both factors had been linked to increases in TILs. To address this question, we have performed multivariate analyses that evaluated independent association between TIL factor vs. the response variable cohort status (SMC, the Korean cohort vs. TCGA, the American cohort) while incorporating multiple variables including molecular subtypes and *BRCA1/BRCA2* mutation status into the predictive model. If the analysis finds that a factor has significant independent association with cohort status, then it is likely a biologically meaningful difference between two cohorts rather than a “bystander” or confounding effect of another factor.

Previously, we have shown in Supplementary Figure 7 that TIL factor (F9) was significantly associated with cohort status after adjusting for confounding effects of key clinical variables including patient age, menopausal status and molecular subtype. In the current version, Supplementary Figure 7 was completely redone and moved to main Figure 5e. As shown in Figure 5e, TIL factor was significantly associated with molecular subtype as it was enriched in ER-negative subtypes (TNBC, HER2+) compared to ER-positive subtypes (ER+, ER+/HER2+) (Figure 5a). However, TIL factor was not associated with patient age or menopausal status after excluding confounding effects of molecular subtype and other clinical factors, indicating that TIL level was not independently associated with patient age. On the other hand, TIL factor remained significantly associated with cohort status ($p = 3.5e-4$) after adjusting for clinical factors. Hence, TIL enrichment in SMC vs. TCGA cannot be fully explained by differences in molecular subtype distribution or age related factors between two cohorts. This point was also illustrated by Figures 5a-b showing higher TIL factor weight and PD-L1 expression in SMC vs. TCGA within HER2+ and TNBC subtypes.

We also performed additional multivariate analysis using the elastic net method that selects independently predictive features for cohort status from a combined list of distinctive molecular and clinical features including molecular subtype, *BRCA1/BRCA2* mutation status and patient age. As shown in Figure 5e, TIL factor (F9) was identified as a more robust predictor than *BRCA1/BRCA2* mutation status among other molecular features. *BRCA1/BRCA2* mutation status was significantly associated with patient age and molecular subtype, but was no longer independently associated with cohort status after excluding these confounders. In another word, the prevalence of *BRCA1/BRCA2* germline “pathogenic” mutation did not significantly differ between SMC vs. TCGA patients within the same age groups (Supplementary Figure 3b). Taken together, these analysis results demonstrated that TIL enrichment in SMC vs. TCGA tumors was unlikely to result from differences in patient age, molecular subtype distribution or *BRCA1/BRCA2* mutation prevalence. The significant independent association between TIL factor and cohort status suggested that this was an inherent difference between Korean vs. American breast cancers.

We have made the following changes to the manuscript:

- Extensively revised RESULTS section “Multivariate analyses of distinctive molecular and clinical features”.
- Added Figure 5e to present the results of regression and EN analyses. Statistical independent associations between molecular features (rows) and clinical features (columns) were shown as a table. Variable usage determined by elastic net were shown as a bar plot.
 - Previous Supplementary Figure 7 was removed.
 - To make space, we moved previous Figures 5c-d to Supplementary Figure 7.
- Added Supplementary Table 13.
- Added method descriptions for the elastic net and regression analyses in Methods section “Multivariate analyses of clinical and molecular features”.

(2) Also, Korean women diagnosed with breast cancer, majority are postmenopausal based on national registry data [5-7], therefore the cohort on patients in this study is not truly representative of Korean BC and the conclusion may be too aggressively stated.

We thank the reviewer for pointing out this important issue. We agree with the reviewer that this SMC cohort is younger in age than Asian/Korean BCs in general and therefore not representative of the age distribution for Asian BCs overall. We would like to clarify that the goal of this study was not to compare all Asian BCs vs. Western BCs, but to characterize the genomic and molecular profile for an important subset of Asian BC patients who are younger and pre-menopausal. This patient segment makes up a significant portion of Asian BCs but is under-represented in western BCs. As discussed in the INTRODUCTION, the molecular bases for this distinctive BC segment remained poorly characterized because previous initiatives such as TCGA and METABRIC mainly focused on western BCs. Hence, towards the goals of elucidating the molecular bases and improving treatment outcomes for all Asian BCs, we feel that our study provides an important first step to delineate the genomic/molecular landscape of younger Asian BCs.

The comprehensive multi-omics dataset from the TCGA BRCA study provided a valuable reference to the scientific community for studying the molecular bases of BCs. Comparison analyses of genomic and molecular profiles between SMC and TCGA served two purposes. First, it allowed us to place the newly generated SMC data into the context of what people already knew about the molecular bases of BC and identify potentially novel and meaningful insights into the disease biology of younger Asian BCs. Second, it shows that there exist significant molecular differences between younger Asian BCs

and the reference BC molecular profiles, thus revealing gaps in the existing data/knowledge and emphasizing the need to perform thorough molecular characterization of BCs from different racial backgrounds and age strata. After the identification of significant molecular differences between two large cohorts that differed in ethnicity, age among other factors, we sought to decipher which factors were causal of the observed molecular distinctions. We believed it would be scientifically significant to identify molecular features associated with younger BCs as well as features associated with Asian BCs, as both patient segments remained poorly understood and under-represented in the currently available molecular data. Towards this end, we have applied rigorous multivariate analysis approaches to deconvolute the complex associations among key clinicopathologic factors and distinctive molecular features including enrichment of TILs, increased HRD in TNBC and down-regulated TGF- β and ER signaling. Innovative analyses of virtual microdissection and integration with DE analyses further revealed tissue compartmental origins for these distinctive molecular features. We found that tumor microenvironment associated features such as TILs and TGF- β signaling were more strongly associated with cohort status than with patient age. On the other hand, tumor intrinsic molecular features such as *BRCA1/BRCA2* mutations and ER signaling were independently associated with patient age/menopausal status rather than with cohort status.

In summary, the main goal of this study was to study the molecular bases for younger Asian BCs, an important but poorly characterized patient segment distinguished by patient age and ethnicity from the established molecular profiles of BCs. By comparing the genomic and molecular profiles of SMC with TCGA, we found several distinctive molecular features of younger Asian BCs that hold potential clinical implications. Further, through applying rigorous and innovative analysis approaches, we have “dissected” the tumor tissue compartment origins and identified likely causal factors for these distinctive features.

Reviewer’s comment indicated to us that the previous draft was not sufficiently clear in explaining these points. We have made the following revisions to the manuscript to address this issue:

- We have revised the last paragraph in INTRODUCTION to emphasize that our goal was to study younger Asian BC.
- We have revised the first paragraph in DISCUSSION to explain the rationale and approach of this study. We also explicitly pointed out the limitations of this study in that our cohort is biased towards younger patients and therefore do not represent overall Asian BC population.
- We have revised ABSTRACT to discuss using multivariate analyses to separate the causal effects of patient age from cohort status i.e. Korean vs. American BC.
- We also revised various parts in the manuscript to avoid claiming that observed differences exist between all Asian BCs and western BCs. We noted that molecular differences were observed between two datasets, SMC vs. TCGA, and potentially represent distinctive molecular features of younger Asian BC populations. Only by applying multivariate analyses were we able to make inferences about which molecular distinctions may be specific to Asian BC in general.

Overall, the data in the article is very interesting and identifying a significant population with DNA repair signature which might benefit from other treatment modalities like PARP inhibitors or platin containing chemotherapy is important for treatment considerations and clinical trial designs. Also, identifying immune activation signature certainly may lead to novel therapy choices, identifying better biomarkers for response to immune mediated therapies and potentially pathways for resistance to these therapies.

Are they novel and will they be of interest to others in the community and the wider field?

The authors build on other studies and do a much more comprehensive study of both clinical, molecular and genomic differences in their patient populations.

Is the work convincing, and if not, what further evidence would be required to strengthen the conclusions?

(3) It is difficult to assess the strength of the conclusion as the statistical methods for comparing of the SMA and TCGA are not described in great detail. But to truly show that Asian/Korean BCs are different from western BCs we would require data from a more representative cohort that included more postmenopausal women.

To address reviewer's concern about method details, we have revised the METHODS section "Multivariate analyses of clinical and molecular features" to provide more details about the statistical methods used for comparison analyses. To identify distinctive molecular features between SMC vs. TCGA that are binary variables e.g. mutation and CNV, we used logistic regression ("lm" method in R) where the group status e.g. cohort (SMC vs. TCGA) was set as the response variable, the molecular feature tested set as the main factor while confounder variables such as molecular subtype were set as covariates. The resulting p-value reflects the association between the main factor and the response variable after adjusting for confounder variables. Benjamini-Hochberg method was used to adjust for multiple hypothesis testing. Linear regression method (the R "limma" package) was used for differential gene expression analysis. Elastic net ("glm" method in R) was used to identify independently predictive features for cohort status (response variable) from a combined list of molecular and clinical features (covariates) that were identified as being differentially distributed between SMC and TCGA.

On a more subjective note, do you feel that the paper will influence thinking in the field?

The results of this paper might influence clinical trial designs to allow for stratification based on genomic/immune subtypes which may correlate with differential responses to therapies.

1. Elesawy, B.H., et al., Immunohistochemistry-based subtyping of breast carcinoma in Egyptian women: a clinicopathologic study on 125 patients. *Ann Diagn Pathol*, 2014. 18(1): p. 21-6.
2. Ihemelandu, C.U., et al., Molecular breast cancer subtypes in premenopausal and postmenopausal African-American women: age-specific prevalence and survival. *J Surg Res*, 2007. 143(1): p. 109-18.
3. Mahmoud, S.M., et al., Tumor-infiltrating CD8+ lymphocytes predict clinical outcome in breast cancer. *J Clin Oncol*, 2011. 29(15): p. 1949-55.
4. Nolan, E., et al., Combined immune checkpoint blockade as a therapeutic strategy for BRCA1-mutated breast cancer. *Sci Transl Med*, 2017. 9(393).
5. Min, S.Y., et al., The Basic Facts of Korean Breast Cancer in 2013: Results of a Nationwide Survey and Breast Cancer Registry Database. *J Breast Cancer*, 2016. 19(1): p. 1-7.
6. Kim, Z., et al., The Basic Facts of Korean Breast Cancer in 2012: Results from a Nationwide Survey and Breast Cancer Registry Database. *J Breast Cancer*, 2015. 18(2): p. 103-11.
7. Kim, Z., et al., The basic facts of korean breast cancer in 2011: results of a nationwide survey and breast cancer registry database. *J Breast Cancer*, 2014. 17(2): p. 99-106.

Reviewer #3 (Remarks to the Author):

This study by Kan and colleagues describes a genomic (exome/transcriptome) study of Korean breast cancer patients (178 cases), of which the majority (88%) are 'young' (pre-menopausal). Comparisons between the Korean cases and TCGA cases (which are largely post-menopausal) revealed differences in the prevalence of certain molecular subtypes and driver genes, such as higher HER2+, Luminal B, and p53 rates in the Korean cases. Notably, germline mutations in BRCA1 and BRCA2 were higher in the Korean population (10% vs 6.4%), and Korean cases had higher mutation signature proportions of homologous recombination deficiency (HRD). Expression based "virtual tissue decomposition" revealed higher levels of immune signatures in Korean populations, suggestive of a more immune-active environment. This study is of interest particularly because a significant number of breast cancer cases in Asia are younger in nature.

Major Comments

(1) A major component of this manuscript involves comparison the profiles of the Korean tumors to TCGA. In order for robust comparisons, ideally both data sets should be remapped according to the same pipeline. From the Methods, it appears that there are differences in the data pre-processing (eg Firehose for TCGA, Varscan for Korea). The analysis should be repeated using a common pipeline.

The reviewer raised an important caveat. To address this concern, we have re-processed the raw WES data for the TCGA BRCA cohort using the same pipeline that analyzed the WES data from SMC. That included calling of somatic, germline mutations and copy number variations. We then updated the following analyses using this new set of TCGA mutations and CNVs: (1) BRCA1/2 germline mutations; (2) significant mutated genes; (3) comparing the landscape of genomic alterations; (4) mutation signature analyses; (5) molecular feature analyses. These updates produced only minor changes in analysis results such as gene-level mutation and alteration frequencies. We noted that a few genes previously reported to harbor higher mutation frequencies in SMC vs. TCGA at marginal significance were no longer significant (e.g. GATA3). Important molecular differences including *TP53* mutation, *ERBB2* amplification, *BRCA1/BRCA2* germline "pathogenic" mutation and HRD signature S3, remained significant after updating the TCGA data, indicating that these results are robust.

We have made the following changes to the manuscript:

- **Revised Figures 2a-c, 3a-f, 5e, Supplementary Figures 2b, 3 and 4.**
- **Revised Table 2, Supplementary Tables S4, S6, S7 and S8.**
- **Revised RESULTS sections**
 - **"Germline pathogenic mutations in BC predisposition genes"**
 - **"Significantly mutated genes"**
 - **"Comparing the landscape of genomic alterations"**
 - **"Identification and comparison of mutation signatures"**

(2) Many of the differences between the Korean and the TCGA cases may be inter-related. For example, if there are more ER negative cases in the Korean set, then this may explain the higher frequency of TP53 mutations in the Korean set, and maybe even the differences in the microenvironment (eg more TP53 mutations -> more mutations -> more neoantigens -> more immune cells etc). I would like to see the authors attempt a multi-variate analysis that takes into account all the individual differences seen (eg age, immune cells etc) that are significant at the individual factor level, and assess which remain significant after multi-variate adjustment.

We thank the reviewer for a very insightful suggestion that would surely improve this manuscript. In the submitted draft version, we have performed regression analyses that evaluated independent

associations between distinctive molecular features (e.g. TIL F9, *ESR1* expression) and key clinical features (e.g. age, tumor stage). To fully address reviewer's comment, we have expanded the scope of that analysis and performed additional multivariate analyses.

Elastic net (EN) is a regularized regression method that uses penalization in a generalized linear model. It combines ridge regression and the lasso method for variable selection that tends to select one variable from a group of highly correlated variables. First we applied EN to identify independently predictive features for different cohorts (SMC vs. TCGA) from a combined list of key clinical features and distinctive molecular features (see Methods "Multivariate analyses to identify independent associations"). Consistent lasso estimation and variable selection was achieved through bootstraps, which allowed us to estimate the % variable usage for each feature based on the frequency of variable selection from 500 bootstrap runs (Figure 5e). In addition, the statistical significance of association between each feature vs. cohort status was determined using a generalized linear model that evaluated the feature of interest along with other clinical and molecular features as covariates (Supplementary Table 13). We also expanded the regression analysis by adding tumor purity to the clinical features and TGF- β signature to the molecular features. The figure was completely redone and moved from a Supplementary Figure to the Main Figure to emphasize its importance.

For the specific example mentioned by the reviewer, *TP53* mutation was indeed associated with molecular subtype and histology subtype (lobular carcinoma) but remained significantly enriched in SMC vs. TCGA ($p = 0.035$) after adjusting for confounders. In addition, Figure 2b showed higher *TP53* mutation prevalence in SMC than in TCGA within multiple subtypes and after excluding lobular carcinoma cases, indicating that observed difference in *TP53* mutation frequency was not entirely confounded by differences in the distribution of molecular and histology subtypes. Figure 5e also shows that TME differences, TIL factor F9 and TGF- β signature, remained very strongly associated with cohort status after correcting for confounding effects of clinical features such as molecular subtype and tumor purity. Further, we did not observe significantly higher mutation burden in SMC vs. TCGA. Hence, it is unlikely that differences in *TP53* mutation frequency or subtype distribution are the main contributory factors to TME or TIL differences between SMC vs. TCGA.

We have made the following changes to the manuscript:

- Extensively revised RESULTS section "Multivariate analyses of distinctive molecular and clinical features".
- Added Figure 5e to present the results of regression and EN analyses. Statistical independent associations between molecular features (rows) and clinical features (columns) were shown as a table. Variable usage determined by elastic net were shown as a bar plot.
 - Previous Supplementary Figure 7 was removed.
 - To make space, we moved previous Figures 5c-d to Supplementary Figure 7.
- Added Supplementary Table 13.
- Added method descriptions for the elastic net and regression analyses in Methods section "Multivariate analyses of clinical and molecular features".

(3) For the cases with germline *BRCA1/2* mutations, how many experienced a second inactivating hit in the tumor (eg LOH, second mutation, silencing which could be investigated by looking at loss of expression, etc).

The reviewer raised a very good point. For previously detected *BRCA1/BRCA2* germline pathogenic mutations, we looked for a "second hit" in the form of LOH (loss-of-heterozygosity), LOE (loss-of-expression: TPM < 1) or another mutation affecting the same gene/tumor. We found evidence of a

second inactivating hit in 60% (12/20) of *BRCA1/BRCA2* mutations with 5 cases harboring LOH, 4 harboring LOE and 3 harboring both LOH and LOE. We have added Supplementary Table 4 to provide detailed data on *BRCA1/BRCA2* germline mutations identified in the SMC cohort.

Biallelic inactivation is the hallmark of *BRCA1* or *BRCA2* loss-of-function (LOF) mutations that cause deficiency in homologous recombination repair (HRD). We recognize that not all of these “pathogenic” germline mutations result in loss-of-function for *BRCA1* or *BRCA2*. On the other hand, these criteria will also miss *BRCA1/BRCA2* missense or in-frame mutations that do result in loss-of-function and HRD. Our goal is to apply relatively simple criteria to enrich for LOF mutations in *BRCA1/BRCA2* for the purpose of comparing between SMC and TCGA.

We have made the following changes to the manuscript:

- Added Supplementary Table 4 to provide SMC’s list of *BRCA1/BRCA2* germline “pathogenic” mutations with annotations including the “second hit status”.

(4) Have the others explored redoing mutation significance analysis on the combined TCGA and Korean cohort? More driver genes may be identified due to the expanded number of samples.

This is a great suggestion. We have performed MutSigCV analyses on three sets of somatic mutations – SMC alone, TCGA alone and combined. There are 109 significantly mutated genes identified by the combined mutation analysis and 84 of these were identified based on TCGA mutations alone (FDR ≤ 10%). No additional gene was identified based on SMC mutations alone. We found 25 genes that were only significant in the combined mutation analysis but none was mutated at above minimum frequency in SMC (> 3 mutated samples). We concluded that we did not identify any additional cancer driver genes by combining the mutation lists.

We have made the following changes to the manuscript:

- Added Supplementary Table 6 to show the 109 significantly mutated genes.
- Added a sentence that describes this analysis to the RESULTS section “Significantly mutated genes”.

(5) Was the F9 factor specific to a particular molecular subtypes of breast cancer (eg ER negative)?

As shown in Figures 4a and 5a, the F9 factor was present in all BC subtypes but enriched in ER-negative subtypes (HER2+, TNBC) compared to the ER-positive subtypes (ER+, ER+/HER2+). This enrichment pattern was observed in both cohorts and consistent with recent clinical trial results that checkpoint inhibitors are more successful in TNBC than in ER+, indicating that ER-positive subtypes are less immunogenic than ER-negative subtypes.

(6) Have the differences in the tumor microenvironment observed in the expression analysis been corrected for potential differences in tumor purity, which is higher in the TCGA cohort? It seems that if one were to compare high-tumor cellularity tumors to low tumor-cellularity cases, then it seems obvious that differences in tumor microenvironment genes will be seen. Is this true biology or an artefact of the sampling procedure?

Thanks for pointing out an important caveat. We have incorporated tumor purity into our DE analyses (SMC-Pre vs. TCGA-Post and SMC-Pre vs. TCGA-Pre) based on multiple regressions that also considered molecular subtype and tumor stage as confounder variables. The DE results were very similar with previous findings that immune/inflammatory, TGF- β signaling and ER signaling pathways constitute the major expression differences between TCGA and SMC tumors. The majority of the DE pathways were still attributed to TME rather than tumor intrinsic compartments.

As shown in Figure 5e, we also added tumor purity into our multivariate analyses that examined independent associations between molecular features and clinical features. That means the confounding effect of tumor purity was excluded along with other clinical features in calculating the independent association between each molecular feature (row) and the clinical feature of interest (column). Tumor microenvironment features including TIL factor F9 and TGF- β signature remained significantly associated with cohort status after correcting for tumor purity and other covariates.

We have made the following changes to the manuscript:

- Revised Supplementary Figures 8-9 and Supplementary Tables 12 to provide the updated DE analysis results.
- Added Figure 5e (previously Supplementary Figure 7) to describe the multivariate analysis results.
- Extensively revised RESULTS section “Multivariate analyses of distinctive molecular and clinical features” (previously titled as “Association of distinctive molecular signatures with clinical features”).

Minor Comments

(1) Table 1 should highlight which clinical factors are significantly difference between the Korea and TCGA cohorts.

This is a great suggestion. We have added a column “Statistical Significance” to Table 1 to provide p-values determined by Student’s t test or Fisher’s exact test for clinical factors that are significantly different from SMC and TCGA.

(2) Are the BRCA1/2 mutations seen in the Korean cases novel, or have any of these been previously reported in other studies (perhaps from Asian breast cancer families)?

All but one of the BRCA1/BRCA2 pathogenic germline mutations in SMC were previously found in the dbSNP database. Please see column titled “dbSNP_match_id” in the newly added Supplementary Table 4.

REVIEWERS' COMMENTS:

Reviewer #2 (Remarks to the Author):

The manuscript is much improved with the revisions, authors address most of the issues brought up by reviewers as allowed by the scope of the study.

Reviewer #3 (Remarks to the Author):

The authors have done a good job in responding to my original concerns. Congratulations on an impressive study.

Reply to reviewers' comments

We thank all three reviewers for favorable responses. We also sincerely thank them for constructive criticisms and helpful suggestions that improved the quality of this work.

Reviewer #2 (Remarks to the Author):

The manuscript is much improved with the revisions, authors address most of the issues brought up by reviewers as allowed by the scope of the study.

Reviewer #3 (Remarks to the Author):

The authors have done a good job in responding to my original concerns. Congratulations on an impressive study.